# The V-ATPase complex component RNAseK is required for lysosomal hydrolase delivery and autophagosome degradation

Agata N. Makar[1], Alina Boraman[1], Peter Mosen [2], Joanne E. Simpson[1], Jair Marques[1], Tim Michelberger[1], Stuart Aitken [3], Ann P. Wheeler[3], Dominic Winter [2], Alex von Kriegsheim [1] & Noor Gammoh [1] ✉

Autophagy is a finely orchestrated process required for the lysosomal degradation of cytosolic components. The final degradation step is essential for clearing autophagic cargo and recycling macromolecules. Using a CRISPR/Cas9-based screen, we identify RNAseK, a highly conserved transmembrane protein, as a regulator of autophagosome degradation. Analyses of RNAseK knockout cells reveal that, while autophagosome maturation is intact, cargo degradation is severely disrupted. Importantly, lysosomal protease activity and acidification remain intact in the absence of RNAseK suggesting a specificity to autolysosome degradation. Analyses of lysosome fractions show reduced levels of a subset of hydrolases in the absence of RNAseK. Of these, the knockdown of PLD3 leads to a defect in autophagosome clearance. Furthermore, the lysosomal fraction of RNAseK-depleted cells exhibits an accumulation of the ESCRT-III complex component, VPS4a, which is required for the lysosomal targeting of PLD3. Altogether, here we identify a lysosomal hydrolase delivery pathway required for efficient autolysosome degradation.

Lysosomes are acidic organelles containing over 60 hydrolases making them major proteolytic compartments in mammalian cells[1]. Precursors of lysosomal hydrolases are synthesised in the ER and subsequently transported to lysosomes where they are activated[2]. The luminal acidic environment of lysosomes ensures a contained activity of the hydrolases. Delivery of cargo to lysosomes can occur via multiple mechanisms including the endocytic pathway and autophagy. One form of autophagy known as macroautophagy delivers cellular cargo via the formation of double-membrane vesicles, called autophagosomes, which can readily fuse with the lysosomes to form autolysosomes[3,4].

Autolysosome degradation is vital for the completion of macroautophagy (hereafter referred to as autophagy). Lysosomal enzymes selectively degrade the inner autophagosomal membrane (IAM), while the outer autophagosomal membrane (OAM) remains intact[5]. IAM disintegration is crucial to expose the autophagosome lumen and subsequent digestion of engulfed cargo, whereas OAM protection is important for the containment of hydrolases within autolysosomes. Immunofluorescence analyses of autolysosomes revealed a formation of a transient acidified ring between the IAM and OAM, suggesting the separation of the two membranes[5]. In yeast, the integral vacuolar membrane protein Atg15/Cvt17 has been identified as an essential phospholipase required for autophagy and the disintegration of autophagic bodies within the vacuole[6]. The specific enzyme(s) responsible for IAM degradation in mammalian cells have not yet been discovered.

PLD3 is a member of the PLD phospholipase family. Unlike PLD1 and PLD2, the phospholipase activity of PLD3 is debatable despite the presence of two HKD domains in the protein[7,8]. Interestingly, PLD3 and Atg15 are functionally similar as both proteins contain an N-terminal transmembrane region and are transported to lysosomes via MVBs

[1]Cancer Research UK Scotland Centre, Institute of Genetics and Cancer, University of Edinburgh, Crewe Road South, Edinburgh, UK. [2]Institute for Biochemistry and Molecular Biology, Medical Faculty, University of Bonn, Bonn, Germany. [3]MRC Human Genetics Unit, Institute of Genetics and Cancer, Crewe Road South, University of Edinburgh, Edinburgh, UK. ✉e-mail: noor.gammoh@ed.ac.uk

through an unconventional mechanism requiring VPS4 activity[9,10]. Recent studies suggest that Alzheimer's disease-associated mutations in PLD3 abrogate its putative phospholipase activity[7,11] and modulating PLD3 levels has been shown to cause an enlargement of endolysosomal vesicles[12,13]. PLD3 was also shown to harbour an exonuclease activity[14] and the degradation of defective mitochondria during mitophagy is disrupted in PLD3 knockout cells[15].

RNAse kappa (RNAseK) is a small protein consisting of two transmembrane domains with the N- and C-termini of the protein predicted to face the cytoplasm[16]. The protein was initially described as an endoribonuclease with an ability to hydrolyse ApU, ApG, and UpU bonds in vitro[17,18]. This enzymatic activity of RNAseK however has not been validated in cells. Subsequent studies identified RNAseK as a crucial factor required for the endocytic uptake of multiple cargoes including viruses[16,19]. Recent studies indicate that RNAseK is the mammalian homologue of the *S. cerevisiae* rotary subunit f (Vma7) which forms part of the V-ATPase proton pump[20]. V-ATPases are ATP-dependent proton pumps operating by a rotary motion and facilitating the unidirectional transport of protons across membranes to maintain the differential pH of cellular organelles[21]. RNAseK and Vma7 display 32% sequence identity and 52% sequence similarity, and structural analyses of mammalian brain V-ATPase revealed RNAseK positioning in the f subunit of the pump $V_o$ region[20]. Despite being part of the V-ATPase pump, its positioning in the complex suggests that its unlikely to play a role in the proton pumping activity[22]. Therefore, the role of RNAseK in cells and its detailed regulatory network remain unclear.

In this study, we performed a CRISPR/Cas9-based genomic screen and identified RNAseK as a crucial factor for autophagy. RNAseK is required for the final stages of autophagosome degradation and plays a role in the proper trafficking and delivery of lysosomal hydrolases.

## Results

### Identification of RNAseK in a screen for autophagy regulators

We made use of a genome-wide sgRNA library[23] to perform a CRISPR-Cas9 knockout screen in GFP-LC3-expressing mouse embryonic fibroblasts (MEFs) and identify genes that regulate autophagy in our cells. Amino acid (AA) starvation is a robust inducer of autophagy that leads to GFP-LC3 degradation in a manner dependent on autophagy-related (ATG) players, such as ATG7 (Fig. 1a). Chemical inhibition of early stages of autophagosome formation (using a VPS34 inhibitor) or lysosomal degradation (using Bafilomycin A1, Baf A1) can also rescue GFP-LC3 degradation in these cells (Fig. 1a). Lentiviruses were used to package the small guide RNA (sgRNA) library targeting genome-wide mouse sequences and infect MEFs stably expressing GFP-LC3 and the Cas9 endonuclease. Subsequently, cells were AA starved and FACS sorted to separate GFP-positive (autophagy deficient) and GFP-negative (autophagy intact) populations (Fig. 1b). Following DNA extraction and sgRNA amplification, Illumina sequencing was used to identify targeted genes crucial for GFP-LC3 degradation, and therefore autophagy, in GFP-positive cells (Fig. 1b). Known *Atg* genes are amongst the top hits therefore confirming the validity of the screen (Fig. 1c, Supplementary Data 1 and Supplementary Data 2). Also amongst the top hits is *Rnasek*.

To characterise the role of RNAseK during autophagy, we used sgRNA sequences (sgRNAseK) to target its expression in MEF cells (Supplementary Fig. 1a). We then assessed key autophagy markers, including LC3 and p62, and observed that cells lacking RNAseK showed an increase in lipidated LC3, LC3-II, and p62 levels under basal conditions (Fig. 1d–f). Whilst both LC3-II and p62 can be targeted for degradation during AA starvation in control cells, the levels of these autophagy players remained unchanged during starvation or Baf A1 treatment in RNAseK knockout cells (Fig. 1d–f). Knockout of RNAseK in the human osteosarcoma cells, U2OS, also resulted in LC3-II accumulation and disrupted degradation (Supplementary Fig. 1b). Similarly, we virally introduced an sgRNA to inhibit the expression of RNAseK in

interscapular brown adipose tissue (iBAT) in mice, which can be efficiently transduced using AAV-based vectors (Fig. 1g), and observed an increase in LC3-II and p62 levels in vivo (Fig. 1h, i). These results suggest that RNAseK is required for autophagy in cultured cells and in animals.

### RNAseK knockout results in the accumulation of autolysosomal structures

To further determine the stage at which autophagy is disrupted in the absence of RNAseK, we then investigated whether the observed defect in LC3-II and p62 degradation was a result of a disruption in autophagosome closure. To do so, we used a proteinase K (PK) protection assay to monitor the accessibility of autophagosome cargo for proteolytic cleavage, where increased susceptibility to PK degradation indicates the accumulation of unclosed autophagosomes[24] (Supplementary Fig. 2a). In this assay, the proteolytic degradation of p62 was comparable between control and RNAseK knockout cells (Supplementary Fig. 2b, c), suggesting that autophagosome closure is intact in the absence of RNAseK. Subsequently, we aimed to test whether the accumulation of LC3-II and p62 in cells was a result of a defect in autophagosome-lysosome fusion in RNAseK knockout cells. Immunofluorescence analyses of GFP-LC3 or p62 and the lysosomal marker LAMP1 showed an accumulation of both proteins within LAMP1 structures in the absence of RNAseK (Fig. 2a, b) suggesting that autophagosome-lysosome fusion is intact. This was confirmed by electron microscopy (EM) where an accumulation of autolysosome structures was observed in cells lacking RNAseK (Fig. 2c).

The above results led us to test whether the degradation of the IAM after lysosome fusion was disrupted in the absence of RNAseK. To test this, we utilised a previously described method to study IAM degradation by colocalising Syntaxin17 (STX17) and LysoTracker[5]. LysoTracker rings positive for STX17 correspond to intact IAM while STX17-positive lysotracker dots (diffused staining) correspond to degraded IAM (Fig. 2d). Interestingly, cells lacking RNAseK exhibited an increase in LysoTracker rings positive for STX17 and a decrease in STX17 positive lysotracker dots when compared to control cells thereby suggesting a disruption in IAM degradation (Fig. 2e). Our findings to this point show that RNAseK is dispensable for autophagosome closure, lysosome fusion, and general lysosome activity but is required for IAM degradation within autolysosomes and thereby autophagic flux.

### RNAseK knockout does not affect general lysosomal function

As RNAseK was recently identified as a subunit of the lysosomal V-ATPase proton pump[20] and plays a role in viral endocytosis[16], we aimed to test whether the effects of RNAseK knockout on autophagy are mainly due to a disruption in lysosomal acidification or degradation. Lysosome acidification can be assessed using the lysosomal dye, LysoSensor Green, which fluorescence intensity correlates with lysosomal acidification[25] and can detect a decrease in lysosomal pH during AA starvation. As can be seen in Fig. 3a, LysoSensor fluorescence was comparable between control and RNAseK knockout cells under both basal growth conditions and during AA starvation, as shown previously[16]. This is in line with structural data showing that subunit f of the V-ATPase (subsequently identified as RNAseK) is not required for the proton pumping activity of the complex[22]. We further confirmed that the lysosomal pH remained acidic in the absence of RNAseK by analysing a ratiometric pH dye, Oregon Green 488, coupled to dextran[26]. Exciting Oregon Green 488-dextran at 440 nm emitted fluorescence independently of pH (pH insensitive). On the other hand, the emission resulting from exciting Oregon Green at 488 nm is sensitive to pH. By incubating cells with Oregon Green 488-

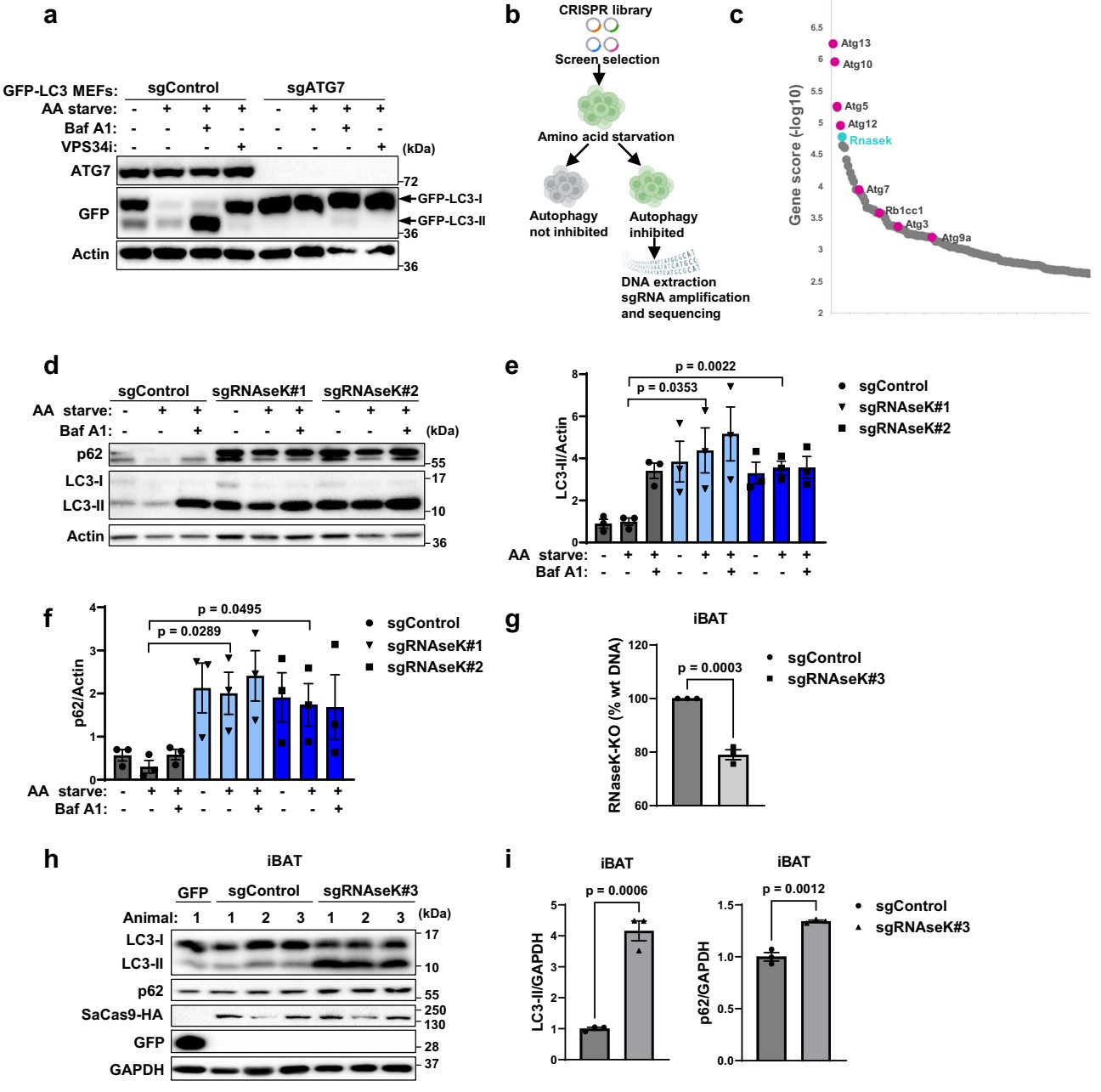

**Fig. 1 | Identification of RNAseK in a screen for autophagy regulators.**
**a** Validation of the GFP-LC3 reporter MEF cell line. Cells expressing Cas9 alone (sgControl) or sgRNA targeting ATG7 (sgATG7) were left untreated or amino acid (AA) starved for 2 h in the presence or absence of Baf A1 or VPS34 inhibitor (VPS34i). Cell lysates were subjected to western blot analyses using the indicated antibodies. $N = 1$. **b** Work flow for loss of function screen. MEFs stably expressing GFP-LC3 and Cas9 were transduced with sgRNA library and subjected to AA starvation (16 h) to induce autophagy and FACS sorting into GFP positive (autophagy deficient) and GFP negative (autophagy competent) populations. Genomic extraction and amplification was used to identify sgRNA sequences. **b** was created with BioRender.com released under a Creative Commons Attribution-NonCommercial-NoDerivs 4.0 International license https://creativecommons.org/licenses/by-nc-nd/4.0/deed.en. **c** Deep sequencing analysis of sorted GFP positive cells. Highlighted are top 100 identified hits, including *Atg* genes and *Rnasek*. **d** Western blot analyses

of sgControl MEF cells and RNAseK knockout cells using the indicated antibodies. Cells were left untreated or AA starved for 2 h in the presence or absence of Baf A1. $N = 3$ biologically independent experiments. **e** Quantification of LC3-II levels normalised to actin in (**d**). **f** Quantification of p62 levels normalised to actin in (**d**). **g** Quantitative DNA assessment of *Rnasek* gene editing efficiency in C57BL/6 mice interscapular brown adipose tissue (iBAT). DNA was extracted from iBAT 2 weeks after AAV injection and analysed using online tools. Gene editing in sgRNAseK#3 was normalised to AAV SaCas9 control (sgControl). $N = 3$ animals. **h** Western blot analyses of iBAT derived from C57BL/6 mice 2 weeks after administration with the indicated AAV vectors using the indicated antibodies. $N = 3$ animals per condition. **i** Quantification of LC3-II and p62 protein levels normalised to GAPDH, as shown in (**h**). In all panels, mean + SEM is assessed by unpaired two-tailed Student's *t* test. Source data are provided with this paper.

dextran for a prolonged time, its accumulation in lysosomes can be used to measure lysosomal pH. Using this method, we observed that lysosomal pH was comparable between RNAseK-deficient and control cells and measures around pH 5.5 (Fig. 3b). On the contrary,

treating cells with Baf A1 elevated lysosomal pH. These data suggest that lysosomal pH remains acidic in the absence of RNAseK.

To further analyse general lysosomal activity, we tested the processing of the lysosomal protease Cathepsin B and observed no

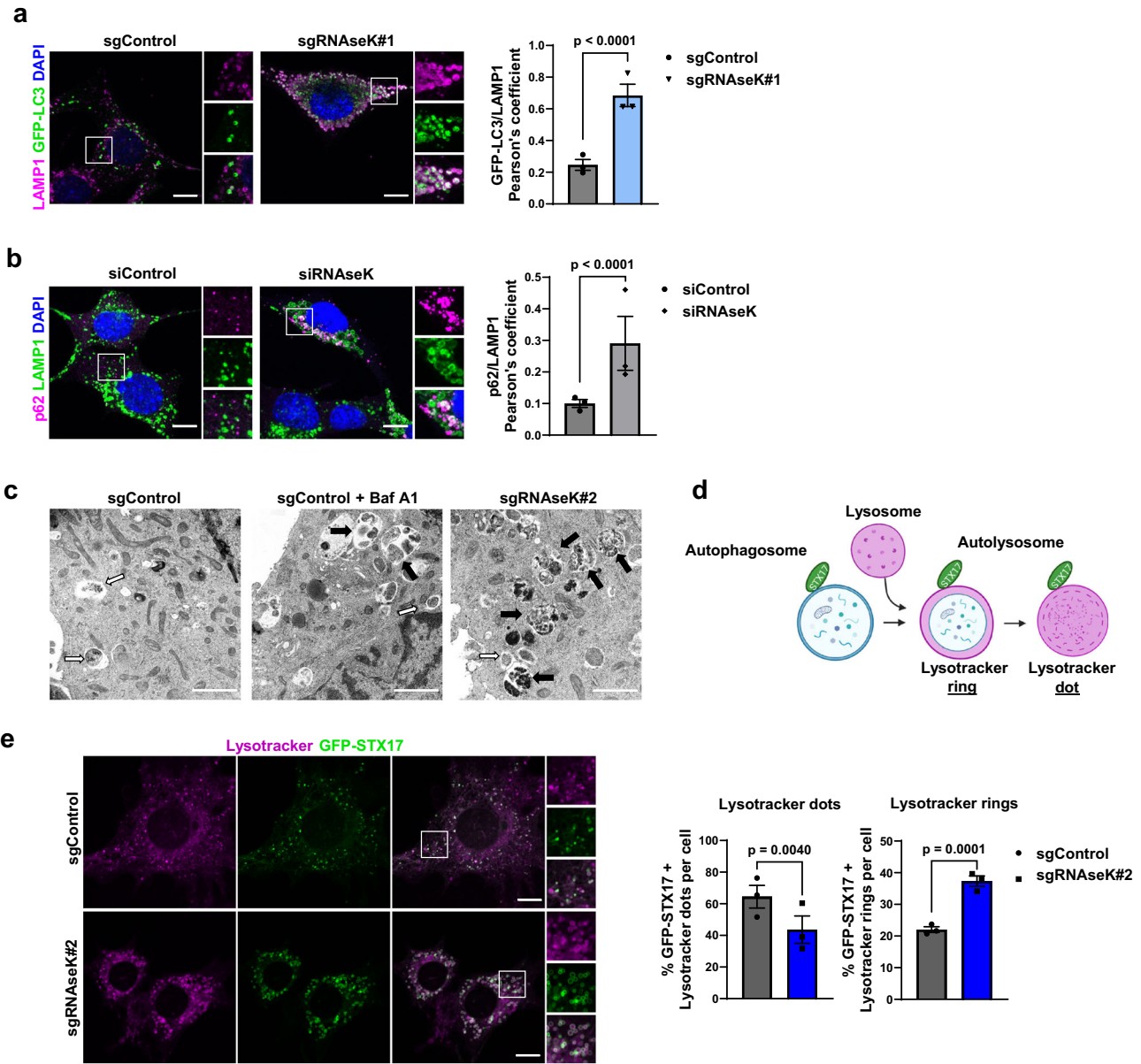

**Fig. 2 | RNAseK knockout results in the accumulation of autolysosomal structures. a** Representative immunofluorescence image of GFP-LC3 and endogenous LAMP1 in sgControl or sgRNAseK MEFs cultured in the absence of AA for 2 h. Scale bar: 10 μm. Quantification of PCC between GFP-LC3 and LAMP1 is shown on the right. $N = 30$ cells from three independent experiments. **b** Representative immunofluorescence image of endogenous p62 and LAMP1 in siControl or siR-NAseK MEFs treated as in (**a**). Scale bar: 10 μm. Quantification of PCC between p62 and LAMP1 is shown on the right. $N = 30$ cells from three independent experiments. **c** Representative electron microscopy images of sgControl and sgRNAseK U2OS cells treated in the absence of AA for 3 h. Baf A1 is added as indicated. Arrows indicate autophagosomes (white arrows) and autolysosomes (black arrows).

Scale bar: 1 μm. $N = 1$. **d** Model of studying IAM degradation whereby STX17 association with undegraded or degraded autolysosomes are observed as a lysotracker ring or dots, respectively. **d** was created with BioRender.com released under a Creative Commons Attribution-NonCommercial-NoDerivs 4.0 International license https://creativecommons.org/licenses/by-nc-nd/4.0/deed.en. **e** Representative images of sgControl and sgRNAseK MEFs stably expressing GFP-STX17 and stained with Lysotracker red. Quantifications of the percentage of STX17 with Lysotracker rings or dots are shown on the right. Scale bar: 10 μm. $N = 15$ cells from three independent experiments. In all panels, mean + SEM is assessed by unpaired two-tailed Student's $t$ test. Source data are provided with this paper.

differences in its proteolytic cleavage, essential for its activity, between control and RNAseK knockout cells (Fig. 3c, d). Cathepsin B activity was further confirmed in RNAseK knockout cells using a fluorogenic substrate, which fluoresces upon its proteolytic cleavage (Fig. 3e, f). Finally, we used EGFR as a lysosomal substrate[27] and showed that EGFR degradation was not affected in RNAseK-depleted cells (Fig. 3g, h). Altogether, these findings suggest that the absence of RNAseK can inhibit autophagy without disrupting overall lysosomal acidification or degradation, consistent with previously published findings[16,20].

## Analyses of the subcellular localisation of RNAseK

To gain a better insight into the specific role of RNAseK during autolysosome degradation, we aimed to test the subcellular localisation of RNAseK. To do so, we generated a MEF cell line where endogenous *Rnasek* was C-terminally tagged with MYC (RNAseK-MYC) to facilitate RNAseK detection by immunofluorescence. To test whether this endogenous tagging at the C-terminal of RNAseK disrupts its role in autophagy, we analysed endogenous LC3 flux and observed no difference between control cells and cells with endogenously tagged RNAseK (Supplementary Fig. 3a). Using these cells, endogenously

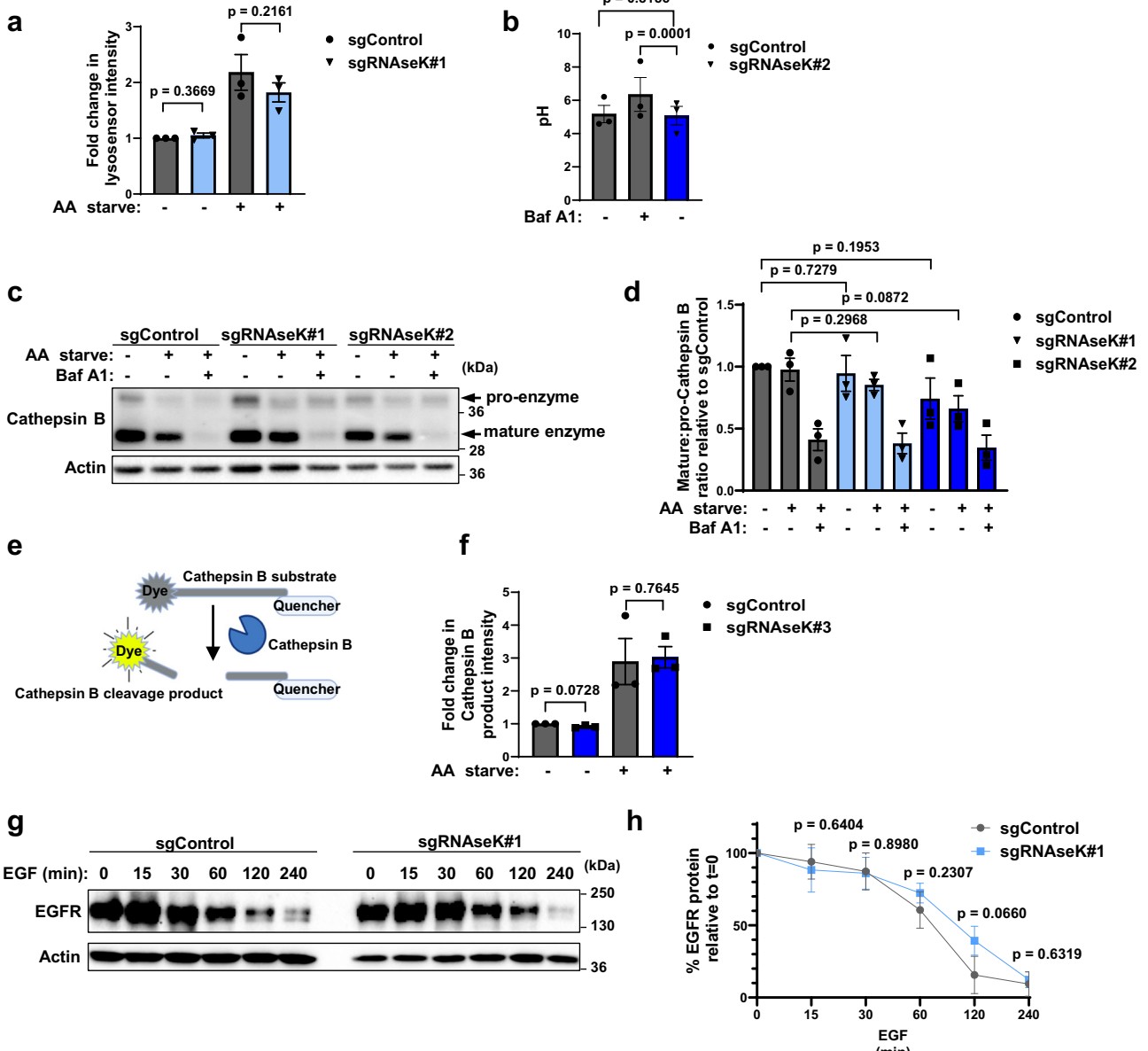

**Fig. 3 | RNAseK knockout does not affect general lysosomal acidification or proteolytic activities. a** Quantification of LysoSensor Green signal fold change in control (sgControl) or RNAseK knockout (sgRNAseK) cells relative to signal in Control cells. MEF cells were left untreated or AA starved for 2 h and incubated with LysoSensor Green for 30 min followed by FACS analyses of fluorescence intensity. Mean + SEM is assessed by paired two-tailed Student's *t* test. *N* = 3 biologically independent experiments. **b** Ratiometric pH quantification using Oregon Green 488-dextran. Control or RNAseK knockout cells were incubated with Oregon Green 488-dextran for 24 h, followed by assessment of fluorescence emission at 520 nm upon excitation at either 440 nm (pH insensitive) or 484 nm (pH sensitive). Calibration using standard buffers was used to determine lysosomal pH. Baf A1 was included as indicated. *N* = 30 cells examined over three independent experiments. **c** Western blot analyses of Cathepsin B in sgControl or sgRNAseK cells. Cells were untreated or AA starved in the presence or absence of Baf A1. **d** Quantification of mature and pro-Cathepsin B relative to sgControl cells in (**c**). *N* = 3 biologically independent experiments. **e** Schematic diagram of Cathepsin B activity assay where

cleavage of Cathepsin B substrate results in the fluorescence of the cleaved product. **e** was created with BioRender.com released under a Creative Commons Attribution-NonCommercial-NoDerivs 4.0 International license https://creativecommons.org/licenses/by-nc-nd/4.0/deed.en. **f** Quantification of Cathepsin B substrate fluorescence in the indicated cells. Signal was normalised relative to sgControl cells. MEF cells were left untreated or AA starved for 2 h and incubated with the Cathepsin B substrate for 30 min followed by FACS analyses. Mean + SEM is assessed by paired two-tailed Student's *t* test. *N* = 3 biologically independent experiments. **g** Western blot analyses of EGFR levels in sgControl and sgRNAseK cells. Cells were cultured without serum for 4 h, followed by stimulation with EGF (20 ng/mL) for the indicated times. *N* = 3 biologically independent experiments. **h** Quantification of EGFR levels following EGF stimulation in (**g**) expressed as a percentage of the EGFR levels at time 0 in the relative cell line. *N* = 3 biologically independent experiments. In all panels, mean + SEM is assessed by unpaired two-tailed Student's *t*-test, unless stated otherwise. Source data are provided with this paper.

tagged RNAseK-MYC was seen to colocalise with various subcellular compartments including lysosomes (LAMP1), late endosomes (RAB7), and Golgi apparatus (GM130) (Fig. 4a–c).

To gain a general understanding of the proximal interaction partners of RNAseK in cells, we expressed RNAseK tagged with a

promiscuous biotin ligase TurboID (TiD) (Supplementary Fig. 3b) and confirmed proper localisation of the tagged protein to late endosomes (Supplementary Fig. 3c). We then performed proximity labelling experiments in MEF cells treated with AA starvation to induced autophagy and exogenous biotin. Biotinylated proteins were then

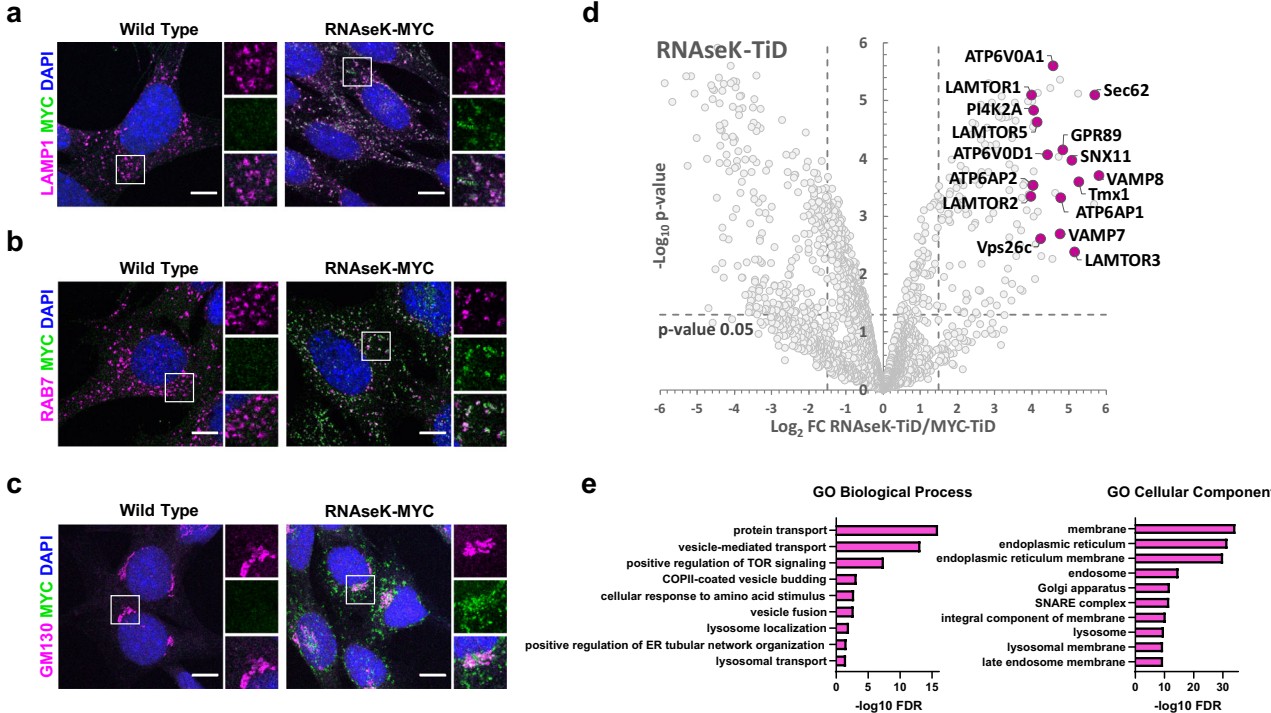

**Fig. 4 | Analyses of the subcellular localisation of RNAseK. a–c** Representative immunofluorescence images of wild type or endogenously tagged RNAseK-MYC MEFs cultured in AA-free media for 2 h. Cells were fixed and stained against MYC and the indicated endogenous proteins. Scale bar: 10 μm. The pearson's colocalisation coefficient (PCC) values for the following colocalisations are: MYC-LAMP1 (PCC = 0.63), MYC-RAB7 (PCC = 0.42), and MYC-GM130 (PCC = 0.22). *N* = 30 cells from three independent experiments. **d** Volcano plot analyses of RNAseK-TurboID (TiD) proximity labelling hits identified by MS presented as relative values to hits obtained from MYC-TiD control cells. Highlighted hits are represented in magenta. The statistical significance was calculated by unpaired two-tailed Student's *t* test. *N* = 1 including three technical repeats. **e** Top ten biological processes (left) and cellular components (right) of hits obtained in (**d**). GO enrichment analysis plotted according to −log₁₀ False Discovery Rate (FDR). The statistical significance was calculated by a one-way Fisher's exact test. Source data are provided with this paper.

purified by streptavidin pulldown and analysed by mass spectrometry (MS). Amongst the identified proximally biotinylated proteins isolated from RNAseK-TiD cells were membrane-embedded members of the V₀ V-ATPase complex domain (Fig. 4d and Supplementary Data 1), which are predicted to lie in close proximity to RNAseK[16]. In addition, gene ontology analyses showed that RNAseK is in close proximity to various endosomal compartments as well as the Golgi apparatus and ER (Fig. 4e and Supplementary Data 1). Altogether, the results from the proximity labelling and imaging experiments suggest that RNAseK closely associates with multiple membranous compartments in cells.

**Lysosomal PLD3 levels are reduced in RNAseK knockout cells**
As RNAseK knockout cells exhibited a defect in autolysosome degradation but not general lysosome function, we speculated whether the delivery of a specific lysosomal hydrolase was altered in the absence of RNAseK. To gain a better overview of lysosomal protein content, we enriched for cellular lysosome fractions using Superparamagnetic Iron Oxide Nanoparticles (SPIONs) followed by magnetic column purification[28]. This method can efficiently enrich for lysosomal proteins (Supplementary Fig. 4a) as shown previously[29]. MS analyses of lysosome-enriched fractions indicated no changes in the abundance of most lysosomal proteases, including Cathepsin B, D, L, and C (Fig. 5a). As expected, lysosome-enriched fractions derived from RNAseK-deficient cells exhibited an increase in autophagy cargo receptors (such as p62, NBR1, and TAX1BP1) consistent with a defect in autophagosome degradation in these cells (Fig. 5a and Supplementary Data 3). Interestingly, a significant decrease was observed in the abundance of a subset of proteins in lysosome-enriched fractions derived from RNAseK

knockout cells compared to their control counterparts. Surprisingly, a significant reduction in the lysosomal level of the V-ATPase complex component ATP6V1D was observed in the absence of RNAseK. However, this decrease in ATP6V1D is not associated with a reduction in lysosomal acidification or dysfunctional lysosomes (Fig. 3). The significance of this reduction in ATP6V1D lysosomal levels remains uncertain but appears to be insufficient to cause disrupted V-ATPase activity.

As RNAseK-deficient cells were associated with a defect in IAM degradation, we were specifically interested in investigating lysosomal lipases decreased in lysosomes derived from RNAseK-deficient cells. These lipases include LIPA, SMPD1, and the putative phospholipase PLD3 (Fig. 5a). To investigate whether inhibiting any of these enzymes affected autophagy, we introduced CRISPR/Cas9-mediated gene editing to inhibit their expression. Interestingly, PLD3 knockout cells resulted in the highest accumulation of LC3 puncta that colocalised within LAMP1-positive endosomes highlighting its role as a potential lysosomal hydrolase required for autophagic flux (Supplementary Fig. 4b). PLD3 is a putative phospholipase with a largely unknown function despite its close association with Alzheimer's disease[30]. The phospholipase activity of PLD3 remains uncertain. Interestingly, PLD3 shares some similarities with the yeast autophagy-related lipase, Atg15, including an N-terminal transmembrane domain and a dependency on the ESCRT-III machinery for delivery to lysosomes[9,10]. To further investigate the reduction in lysosomal PLD3 levels in RNAseK-deficient cells, we analysed cell media to test whether PLD3 is being preferentially secreted. Indeed, an increase in secreted PLD3 was observed in cell media derived from RNAseK knockout cells

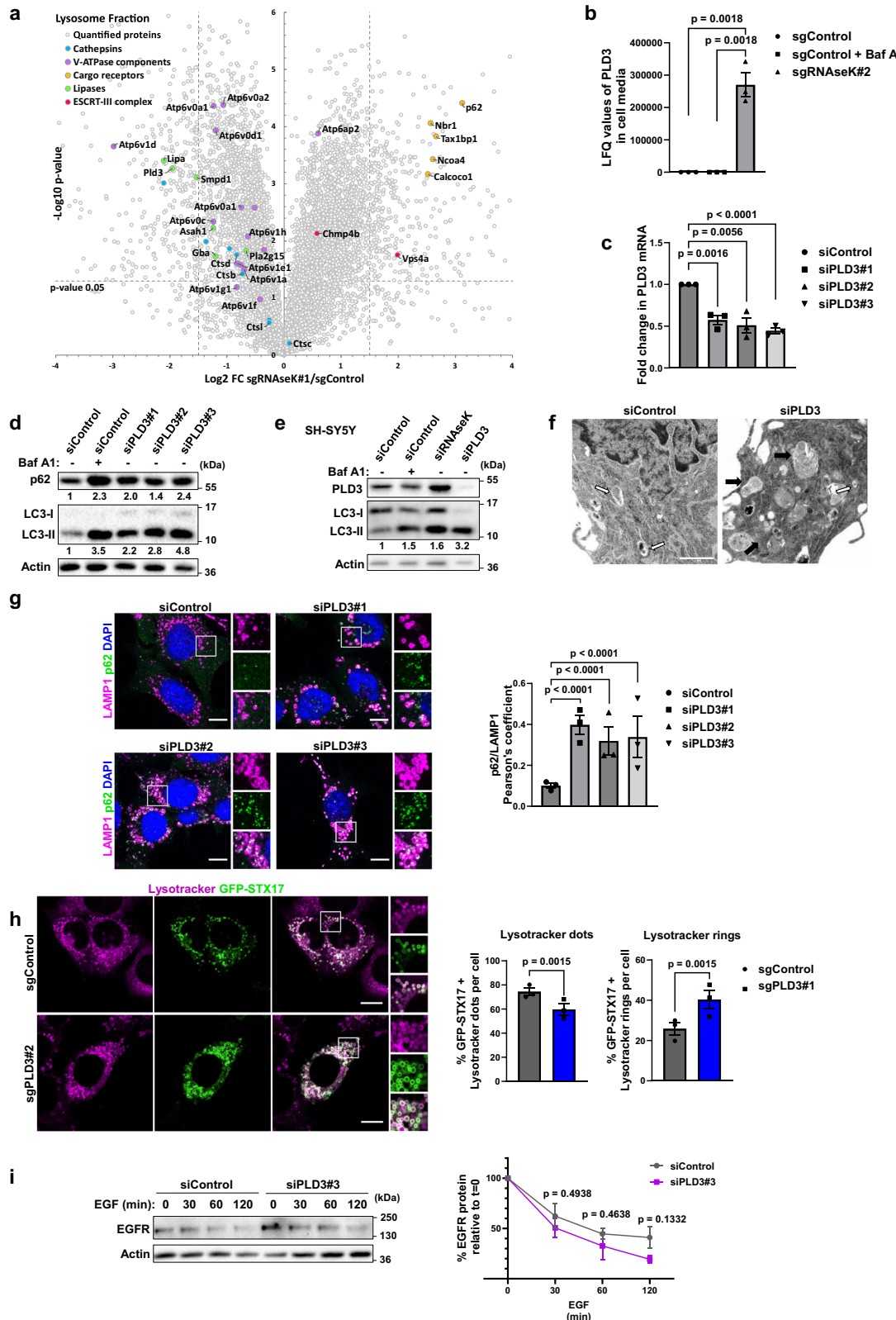

compared to control cells suggesting that PLD3 is differentially trafficked in the absence of RNAseK (Fig. 5b). Inhibiting lysosomal degradation by Baf A1 treatment, previously shown to induce the secretion of lysosomal proteins[31], did not result in enhanced PLD3 secretion suggesting a specificity in PLD3 secretion in the absence of RNAseK (Fig. 5b). Analyses of PLD3 levels in MEFs whole cell lysates showed that its cellular levels are reduced in the absence

of RNAseK reinforcing its enhanced secretion and reduced lysosomal targeting upon RNAseK knockout (Supplementary Fig. 4c). Together, these findings suggest that PLD3 is improperly trafficked in the absence of RNAseK resulting in its reduced lysosomal levels. These findings suggest that PLD3 may be one factor that contributes to the disrupted autophagy phenotype observed in RNAseK-deficient cells.

**Fig. 5 | PLD3 is required for autophagy. a** Quantitative proteomic analysis of enriched lysosomal fractions derived from sgControl or sgRNAseK MEFs. The statistical significance was calculated by unpaired two-tailed Student's t test. **b** MS analyses of PLD3 levels in cell media derived from sgControl or sgRNAseK MEFs. Baf A1 treatment was added as indicated (24 h). Absolute PLD3 intensity is shown as label-free quantitation (LFQ) values. N = 1 including three technical repeats. **c** qRT-PCR analyses of mRNA from MEFs transfected with non-targeting siRNA (siControl) or siPLD3. *Pld3* mRNA levels were normalised to *Actin*. Fold change in expression was compared to siControl cells. N = 3 biologically independent experiments. **d** Western blot analyses of cell lysates derived from siControl MEFs or cells transfected with siRNA sequences targeting PLD3. Baf A1 treatment (2 h) was included as indicated. Quantification of p62 and LC3-II band intensity normalised relative to siControl (lane 1) is shown. N = 3. **e** Western blot analyses of lysates derived from SH-SY5Y cells transfected with siControl or pool siRNAs targeting RNAseK or PLD3. Baf A1 treatment (3 h) was included as indicated. Quantification of LC3-II band intensity normalised relative to siControl (lane 1) is shown. N = 3.

**f** Representative electron microscopy images of siControl and siPLD3 MEFs treated in the absence of AA (3 h). Arrows indicate autophagosomes (white) and autolysosomes (black). Scale bar: 1 μm. N = 1. **g** Representative immunofluorescence images of MEFs transfected with siControl or siPLD3. Cells were cultured in the absence of AA (2 h). Scale bar: 10 μm. Quantification of PCC between p62 and LAMP1 in siControl and siPLD3 is shown on the right. N = 30 cells from three independent experiments. **h** Representative images of sgControl and sgPLD3 MEFs expressing GFP-STX17 and stained with Lysotracker red. Quantifications of the percentage of STX17 with Lysotracker rings or dots are shown on the right. Scale bar: 10 μm. N = 15 cells from three independent experiments. **i** Western blot analyses of EGFR levels in siControl and siPLD3 MEF cells. Cells were cultured without serum (4 h), followed by stimulation with EGF (20 ng/mL). Quantification of EGFR levels as a percentage of time 0 is shown on the right. N = 3 biologically independent experiments. In all panels, mean + SEM is assessed by unpaired two-tailed Student's t test. Source data are provided with this paper.

## PLD3 is required for autophagy

We further investigated the relevance of PLD3 in autophagy by introducing individual siRNA sequences to inhibit its expression (Fig. 5c). siRNA-mediated knockdown of PLD3 in MEF cells resulted in an accumulation of LC3-II and p62 in a manner resembling Baf A1 treatment (Fig. 5d), as observed in RNAseK-deficient cells. Similar results were observed upon PLD3 depletion in SH-SY5Y human neuroblastoma cells (Fig. 5e). Autolysosome accumulation in the absence of PLD3 was also confirmed by EM analyses (Fig. 5f). In addition, the absence of PLD3 led to increased p62 and LC3 puncta that localised within LAMP1-labelled lysosomes (Fig. 5g and Supplementary Fig. 4d) suggesting that autophagy is inhibited post lysosome fusion. Interestingly, lysosomes appeared to be enlarged in the absence of PLD3, as previously reported[12]. Inhibited autolysosome degradation in the absence of PLD3 was further confirmed by analysing lysotracker structures positive for STX17 (as described in Fig. 2d, e) which showed an accumulation of lysotracker ring structures (Fig. 5h) indicating an inhibition in IAM degradation. The observed disruption in autophagosome degradation in PLD3-deficient cells was not a result of a general defect in lysosomes as EGFR degradation was not affected in cells depleted of PLD3 (Fig. 5i). Overall, these results suggest that the absence of PLD3 affects autophagosome clearance but not overall lysosomal degradation in a manner resembling RNAseK inhibition.

## VPS4a accumulates on vesicular membranes in the absence of RNAseK

We further aimed to investigate the altered trafficking of PLD3 in RNAseK knockout cells. Previous studies have shown that PLD3 is targeted to lysosomes via multivesicular bodies (MVBs) requiring the activity of the ESCRT-III component, VPS4a[10]. We therefore tested whether VPS4a activity is affected by RNAseK depletion and thereby may contribute to the defect in PLD3 localisation. Interestingly, our MS analyses of lysosome-enriched fractions derived from RNAseK knockout cells showed increased levels of VPS4a (Fig. 5a), whereas published findings suggest that VPS4a membrane localisation is transient and is found diffused in cells[32]. To further investigate this aberrant localisation of VPS4a in RNAseK-deficient cells, we expressed a GFP-tagged version of VPS4a and observed its diffused localisation in control cells and cells treated with Baf A1 (Fig. 6a), suggesting that inhibiting V-ATPase proton pump activity does not cause changes in VPS4a localisation. Interestingly, when visualised in cells depleted of RNAseK, VPS4a exhibited a dramatic increase in punctate structure formation (Fig. 6a), confirming its aberrant localisation on membranous compartments in the absence of RNAseK. VPS4a puncta in RNAseK-deficient cells showed enhanced colocalisation with the endosomal markers, RAB7 and CD63, as well as p62 (Fig. 6b–d). This altered subcellular localisation of VPS4a was further

confirmed by comparing its interaction partners in control and RNAseK knockout cells using GFP-TRAP pulldown. As can be seen in Fig. 6e, an increased association between VPS4a and RAB7, STX17, and p62 was observed in cells lacking RNAseK further underscoring an altered localisation of this protein in the absence of RNAseK (Supplementary Data 1). Whether these enhanced associations of VPS4a are due to direct protein-protein interactions remains to be tested but it is also possible that they occur due to indirect bindings as a result of increased protein localisation to membranous compartments.

VPS4a forms part of the lysosome damage repair response and is recruited to lysosomes along with Galectin-3, CHMP4b, and Alix in the presence of the lysosomotropic detergent, LLOMe[33,34]. We, therefore, sought to determine whether RNAseK-deficient cells trigger a lysosome damage response and stained cells for markers of lysosomal damage and permeabilization, including Galectin-3 and Alix. As can be seen in Supplementary Figs. 5a–c, Galectin-3 and Alix were recruited to lysosomes in control cells treated with LLOMe but not in RNAseK knockout cells, suggesting that the depletion of RNAseK does not trigger lysosomal perturbations.

We predict that the membrane localisation of VPS4a in RNAseK-deficient cells leads to an inhibition of its activities which perturbs the targeting of PLD3 to lysosomes[10] required for autophagy. Indeed, cells expressing a dominant negative mutant of VPS4a (VPS4a[E228Q]) also resulted in an accumulation of both LC3-II and p62 demonstrating its role during autophagosome clearance (Fig. 6f) as has been previously shown[35]. To further investigate whether VPS4 activity is disrupted in the absence of RNAseK, we analysed an independent cellular activity that mediate plasma membrane repair previously reported to require VPS4 and ESCRT-III components[36] but not the canonical autophagy pathway[37]. This plasma membrane damage can be monitored by measuring propidium iodide (PI) permeability which occurs at higher rates in cells where damage has not been efficiently repaired. Cells lacking RNAseK showed higher PI staining when compared to control cells during digitonin-induced plasma membrane damage suggesting a defect in plasma membrane repair (Supplementary Fig. 5d). Altogether, these results suggest that the subcellular localisation of VPS4a is aberrant in the absence of RNAseK, which may lead to a disruption in some of its activities.

Having observed a role of RNAseK in regulating VPS4a localisation, we tested the possibility of an interaction between the two proteins. While we did not detect RNAseK in VPS4a pulldown (Fig. 6e), this can potentially be due to the fact that RNAseK is a small protein and its proteolytic digest during MS processing precludes its detection. We therefore used transient gene expression and observed that GFP-TRAP pulldown of GFP-RNAseK can co-precipitate mCherry-VPS4a overexpressed in HEK293T (Fig. 6g). Similarly, overexpression of GFP-VPS4a and subsequent GFP-Trap pulldown showed an interaction with

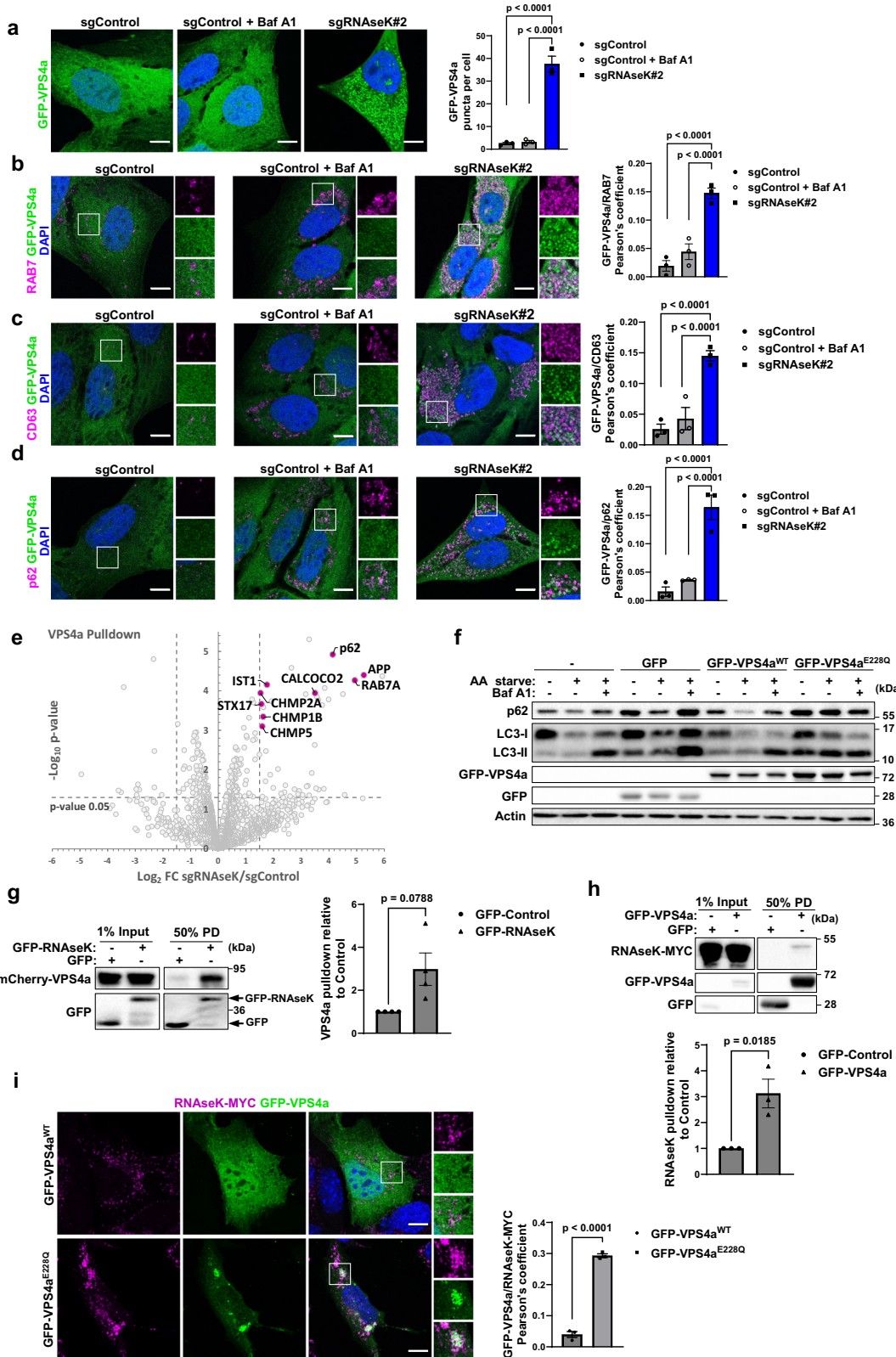

RNAseK-MYC in cells (Fig. 6h). We further assessed the colocalisation of RNAseK with VPS4a$^{E228Q}$, which was shown to be trapped on membranes[32], and observed a significant degree of colocalisation between the two proteins in RNAseK-MYC endogenously tagged cells (Fig. 6i). These findings demonstrate an interaction between RNAseK and VPS4a in cells that can potentially result in the regulation of VPS4a localisation in cells.

## Discussion

Here, we show that the absence of the V-ATPase complex component RNAseK can alter lysosomal protein content independently of changes in lysosomal pH and general proteolytic activity. We show that in cells lacking RNAseK, autophagosome, but not endosome, degradation is disrupted. This is associated with an altered delivery of a number of lysosomal hydrolases of which PLD3 is required for autophagosome

**Fig. 6 | VPS4a accumulates on lysosomal membranes in the absence of RNAseK.**
**a** Representative confocal microscopy images of GFP-VPS4a expressed in sgControl or sgRNAseK U2OS cells and cultured in AA starvation media for 3 h. Baf A1 was added as indicated. Scale bar: 10 μm. Quantification of VPS4a puncta per cell is shown on the right. $N = 30$ cells from three independent experiments.
**b**–**d** Representative immunofluorescence images of GFP-VPS4a expressed in U2OS cells as in (**a**) and stained using antibodies against the indicated endogenous proteins. Scale bar: 10 μm. Quantifications of PCC between GFP-VPS4a$^{WT}$ and the indicated markers are shown on the right. $N = 30$ cells from 3 independent experiments. **e** Volcano plot of interacting proteins identified by MS following GFP-TRAP pulldown of GFP-VPS4a transfected into sgControl or sgRNAseK MEF cells. Selected hits showing significant association with GFP-VPS4a in sgRNAseK cells relative to sgControl cells are highlighted in magenta. The statistical significance was calculated by unpaired two-tailed Student's $t$ test. $N = 1$ including three technical repeats. **f** Western blot analyses of U2OS cells transiently expressing the indicated constructs. Cells were AA starved in the presence or absence of Baf A1 for 3 h. $N = 1$. **g** HEK293T cell lysates transfected with GFP or GFP-RNAseK plasmids were mixed with lysates expressing mCherry-VPS4a followed by GFP-TRAP pull-down (PD) and western blotting analyses using the indicated antibodies. $N = 3$. Quantification of VPS4a pulldown relative to control is shown on the right. **h** HEK293T cell lysates transfected with GFP or GFP-VPS4a plasmids were mixed with lysates expressing RNAseK-MYC-TurboID. Cell lysates were treated as in (**g**). $N = 3$. Quantification of RNAseK pulldown relative to control is shown below. **i** Representative immunofluorescence images of endogenously tagged RNAseK-MYC MEF cells transfected with GFP-VPS4a$^{WT}$ (top panel) or GFP-VPS4a$^{E228Q}$ (bottom panel) and cultured in the absence of AA for 2 h. Cells were fixed and stained using antibodies against MYC. Scale bar: 10 μm. Quantification of PCC between RNAseK-MYC and GFP-VPS4a is shown on the right. $N = 30$ cells from three independent experiments. In all panels, mean + SEM is assessed by unpaired two-tailed Student's $t$ test. Source data are provided with this paper.

degradation. We also show that RNAseK can bind VPS4a, previously demonstrated to be required for the lysosomal delivery of PLD3 via the MVB pathway[10], and in its absence VPS4a predominantly localises to punctate structures. Overall, our findings suggest that RNAseK can facilitate autophagosome degradation by ensuring efficient lysosomal delivery of certain hydrolases including PLD3 and regulating VPS4a localisation in cells (Supplementary Fig. 5e).

Whilst VPS4a usually only transiently localises to membranes, for example during autophagosome closure[38], its predominant membrane localisation in RNAseK-deficient cells may suggest a degree of inhibition of its activity. This may alter some of the functions of VPS4a in cells, such as the trafficking of PLD3 to lysosomes via MVBs[10] and plasma membrane damage repair[36], but not other activities, such as autophagosome closure[35]. How RNAseK regulate VPS4 activity and release from endosomes remains an open question and may involve a degree of regulation of its ATPase activity or post-translational modifications of lysosomal proteins.

RNAseK-deficient cells could distinguish lysosomal degradation of autophagosomes from endosomes. The specific requirement for RNAseK during autophagosome but not endosomal cargo degradation remains to be further analysed and may be a result of different lipid compositions of these vesicles. It is also possible that endosomal membrane proteins are accessible to proteolytic degradation, which appears to remain intact in the lysosomes of RNAseK-deficient cells, thereby destabilising endosomal vesicles. On the other hand, access to proteins within autophagosomes requires an initial degradation of the IAM likely mediated by a lipolytic activity. Our data show that RNAseK is required for the delivery of PLD3 and multiple hydrolases to lysosomes. The specific enzymatic activity which mediates autophagosome degradation remains to be further investigated.

Our findings do not exclude the possibility that multiple phospholipases are involved in autolysosome membrane degradation. Indeed, our proteomic analyses revealed a number of hydrolases to be reduced in the lysosomal fractions isolated from RNAseK-deficient cells. It is possible that the hydrolyses of various lipids that constitute the IAM are mediated by different phospholipases and introducing genetic inhibition of multiple lipases simultaneously may result in more robust effects on autophagosome degradation compared to PLD3 inhibition alone. The functional similarities between PLD3 and Atg15 (discussed above) renders it as an interesting potential phospholipase that mediates IAM degradation. As PLD3 was also ascribed an exonuclease activity[14], its precise catalytic activity required for autophagy remains to be verified. Future detailed explorations are required of the effects of PLD3 on autophagosomal lipid components using cellular or purified in vitro settings. In addition, it would be interesting in future studies to test whether the Alzheimer's disease-associated mutations in PLD3 that disrupt its putative phospholipase activity[7,11] result in a defect in autophagosome degradation. Our results however suggest that optimising PLD3 activity in lysosomes may have a therapeutic potential in restoring autophagy in cells during aging.

## Methods

### Ethics statement
This research complies with all relevant ethical regulations. All animal experiments described in this study were performed in accordance with protocols approved by UK Home Office guidelines.

### Cell culture and treatment
Mouse embryonic fibroblasts (MEFs), U2OS, 293AAV cells (Cell Biolabs; #AAV-100), and HEK293T cells were cultured in Dulbecco's Modified Eagle's Medium (DMEM) containing 4.5 g/L glucose and supplemented with 10% (v/v) Fetal Bovine Serum (FBS), 2 mM L-glutamine, 10 U/mL penicillin, and 100 μg/mL streptomycin (all from Gibco). Human neuroblastoma SH-SY5Y cells were cultured in Eagle's Minimum Essential Medium (EMEM) (Sigma; #M2279) supplemented with 10% (v/v) FBS, 2 mM L-Glutamine, 10 U/mL penicillin and 100 μg/mL streptomycin, 1% (v/v) non-essential amino acids (NEAA) (Gibco; 11140), and 1% (v/v) sodium pyruvate (Gibco; 11360). All cells were maintained in a humidified chamber maintained at 37 °C and 5% $CO_2$.

DMEM lacking amino acids (AA) made in house was used to induce autophagy in cells. The following treatments were used (all dissolved in DMSO): VPS34i (VPS34-IN1, 2 μM; Cayman, 17392); Bafilomycin A1 (Baf A1, 20 nM; Sigma); Leu-Leu methyl ester hydrobromide (LLOMe, 0.5 mM; Sigma-Aldrich L7393).

### siRNA transfections
siRNA-mediated knockdown in MEFs or SH-SY5Y cells was performed using ON-TARGETplus$^{TM}$ duplexes (Dharmacon, Horizon Discovery Biosciences Ltd) targeting mouse *Rnasek* (Gene ID: 52898; L-054958-01-0005), mouse *Pld3* individual siRNAs (Gene ID: 18807, #1-3: J-049108-09-0002; J-049108-10-0002; J-049108-11-0002), human *RNASEK* (Gene ID: 440400;:L-032392-02-0005), and human *PLD3* (Gene ID: 23646; L-009659-01-0005). Control cells were transfected with a non-targeting control siRNA (D-001810-10-05). siRNA duplexes were introduced to cells using Dharmafect 1 transfection reagent (Dharmacon, Horizon Discovery Biosciences Ltd) at a final concentration of 25 nM according to the manufacturer's instructions and cells were assayed 72 h after transfection.

### Transient transfection and viral infection
HEK293T cells were used to generate lentiviral and retroviral particles by transfection with polyethylenimine (PEI). All viruses were filtered through 0.45 μm membranes before infecting target cells in the presence of polybrene. Vectors expressing Cas9 alone (sgControl) was used as a control in all experiments. Transient transfection in U2OS or MEF cells was performed using Lipofectamine 2000 (Invitrogen).

## Plasmids for gene editing and transgene expression

For exogenous expression of RNAseK, PCR amplification from mouse cDNA (NM_173742) sequence (Origene, MR200295) was performed and subcloned as follows. pQCXIP-GFP-RNAseK was cloned using XhoI and EcoRI restriction sites and the following primer sequences: Full length For: 5′-GCAGCACTCGAGATGGCGTCGCTCCTGTGCTGT-3′, Rev: 5′-GCAGCAGAATTCCTAGCGCACCATGTATTCCTT-3′. MYC-TurboID construct was cloned from C1(1-29)-TurboID-V5_pCDNA3 (a gift from Alice Ting, Addgene plasmid #107173) into pQCXIN plasmid by PCR using BamHI and EcoRI restriction sites and the following primer sequences: MYC-TurboID For: 5′-GCAGCAGGATCCATGGAACAAAAACTTATTTCTG AAGAAGATCTGAAAGACAATACTGTGCCTCTG-3′ Rev: 5′-GCAGCA-GAATTCTTACTTTTCGGCAGACCGCAGACT-3′. C-terminally tagged RNAseK-MYC-TurboID pQCXIN vector was generated by PCR using NotI and BamHI restriction sites and the following primer sequences: For: 5′-GCAGCAGCGGCCGCATGGCGTCGCTCCTGTGCTG-3′ and Rev: 5′-GCAGCAGGATCCGCGCACCATGTATTCCTTGC-3′. A linker sequence (containing 3XGGGGS) was then inserted using BamHI restriction site and the following forward primer: 5′- GATCCGGTGGAGGAGGTTCTG-GAGGCGGTGGAAGTGGTGGCGGAGGTAGC-3′.

To generate knockout cells by CRISPR/Cas9-mediated gene editing, lentiCRISPR V2 (a gift from Feng Zhang, Addgene plasmid #52961) was used expressing SpCas9 and single guide RNA (sgRNA) targeting the following mouse sequences: sgRNAseK#1: 5′-CTACAACTGTT TCATCGCCG-3′; sgRNAseK#2: 5′-GTGGCATCGTCCTCAGCGCC-3′ (sequence also used to target human RNAseK); sgPLD3#1 5′-ACTGG-CACCTCGAGGTGTAA-3′; sgPLD3#2 5′-TGCTGTGAGCACCGGCAAGG-3′; *sgLIPA#1 5′-AGTATTCACCGAATCCCTCG-3′; sgSMPD1#1 5′-ATCAA CTCCACAGATCCTGC-3′*. *Atg7* gene knockout was generated using two vector system including lentiCas9-Blast (a gift from Feng Zhang, Addgene plasmid #52962) and lentiGuide-Puro (a gift from Feng Zhang, Addgene plasmid #52963) expressing the following sgRNA targeting mouse *Atg7*: sgATG7: 5′-TCACAGGTCCCCGGATTAG-3′.

For in vivo CRISPR/Cas9-mediated editing of *Rnasek*, pX601-AAV-CMV::NLS-SaCas9-NLS-3xHA-bGHpA;U6::BsaI-sgRNA (a gift from Feng Zhang, Addgene plasmid #61591a) was used containing Cas9 from Staphylococcus aureus (SaCas9) and the following sgRNA sequence targeting *Rnasek*: sgRNAseK#3: 5′-TGGCATCGTCCTCAGCGCCTG-3′. For in vivo interscapular brown adipose tissue (iBAT) targeting, pAAV-GFP (Cell Biolabs; #AAV-400) was used as a transduction control.

For endogenous tagging of *Rnasek* at the C-terminal end with a MYC tag (RNAseK-MYC) in MEF cells, the following constructs were transfected into cells: pcDNA3.3-hCas9 (a gift from George Church, Addgene plasmid #41815), pBabe-GFP (a gift from William Hahn, Addgene plasmid # 10668), a lentiGuide-Puro construct expressing sgRNA targeting RNAseK (5′-ATACATGGTGCGCTAGAGCG-3′), and a donor DNA consisting of mutated STOP codon and PAM site, with MYC tag flanked by ~85 bp homology arms synthesised by Eurofins (5′-GCTACAACTGTTTCATCGCCGCGGGCCTCTACCTCCTCCTCGGAGGC TTCTCCTTCTGCCAAGTTCGTCTCAACAAGCGCAAGGAATACATGGT GCGCgaacaaaaacttatttctgaagaagatctgTAGAGCGCGCTCCGCCTCTC CCTCCCCAGCCCCCTTCTCTATTTAAAGACTCCGCAGACTCCGTCCC ACTCATCTGGCGTCCTTTGGGACTT-3′). 24 h post transfection, cells were selected with puromycin (2 μg/mL) for 72 h followed by single-cell sorting based on GFP fluorescence. Successful tagging in cells was confirmed by sequencing and immunofluorescence staining for MYC tag.

The following constructs were obtained from Addgene: pMRXIP-GFP-STX17 (a gift from Noboru Mizushima, Addgene plasmid #45909), pLNCX2-mCherry-VPS4A (a gift from Sanford Simon, Addgene plasmid #115334), pLNCX2-mEGFP-VPS4A (a gift from Sanford Simon, Addgene plasmid #116924), and pEGFP-VSP4$^{E228Q}$ (a gift from Wesley Sundquist, Addgene plasmid #80351).

## Antibodies

Antibodies against the following targets were used for western blotting (WB), immunofluorescence (IF), and immunoprecipitation (IP), as indicated: β-Actin (WB 1:3000, Clone AC-74, Sigma-Aldrich, A2228); ATG7 (WB 1:2000, Sigma-Aldrich, A2856); Cathepsin B (WB 1:3000, R&D systems, AF965); CD63 (IF 1:300, Clone sc-5275, Santa Cruz, MX-49.129.5); EGFR (WB 1:2000, Clone sc-03-G, Santa Cruz, 1005); GAPDH (WB 1:3000, Clone D16H11, Cell Signaling Technology, CST, 5174); GFP (WB 1:3000, IF 1:300, Chromotek, PABG1-100); GFP (WB 1:3000, IF 1:300, Roche, 11814460001); GM130 (IF 1:300, BD biosciences, 610822); HA-Tag (WB 1:1000, IF 1:300, Clone 3F10, Roche, 11867423001); HA-tag (IF 1:300, Clone C29F4, CST, 3724); LAMP1 (IF 1:1000, Abcam, ab25245, for mouse cells); LAMP1 (IF: 1:500, clone D4O1S, CST, 15665, for human cells); LC3B (WB 1:3000, Sigma-Aldrich, L7543); LC3B (IF 1:200, clone 5F10, Nanotools, 0231-100); MYC-Tag (WB 1:1000, IF 1:300, Clone 9B11, CST, 2276); MYC-Tag (IF 1:300, Clone 71D10, CST, 2278 s); p62 (IF 1:300, Enzo Life Sciences, BML-PW9860-0100); p62 (WB 1:2000, CST, 5114); PLD3 (WB 1:1000, IF 1:200, Atlas Antibodies, HPA012800); RAB7 (IF 1:300, CST, 9364); RFP-tag (WB 1:2000, Rockland, 600-401-379); EGFR (WB 1:1000, CST, 2234); Galectin-3 (IF: 1:100, R&D Systems, AF1197); Ubiquitin (IF 1:300, Upstate Cell Signaling Solutions, 07-375); Alix (IF 1:300, clone 3A9, CST, 2171S); EEA1 (WB 1:3000, Clone C45B10, CST, 3288S); Tubulin (WB 1:3000, CST, 9364); anti-mouse-HRP secondary (WB 1:3000, CST, 7076); anti-rabbit-HRP secondary (WB 1:3000, CST, 7074); anti-goat-HRP secondary (Invitrogen, 61-1620); anti-rabbit-Alexa 488 (IF 1:500, Invitrogen, A11008); anti-rabbit-Alexa 594 (IF 1:500, Invitrogen, A11012); anti-mouse-Alexa 488 (IF 1:500, Invitrogen, A1101); anti-mouse-Alexa 594 (IF 1:500, Invitrogen, A11032); and anti-rat-Alexa 594 (IF 1:500, Invitrogen, A11007); anti-goat-Alexa 594 (IF: 1:500, Invitrogen, A21469); anti-mouse-Alexa 488 (IF: 1:500, Invitrogen, A21200).

Custom antibodies raised against RNAseK were produced by Proteintech using the following antigen peptide sequence: CQVRLNKRKEYMVR (WB 1:2000, IF 1:600).

## Autophagy loss of function screen

MEF cells stably expressing pBabe-GFP-LC3 and SpCas9 (lentiCas9-Blast, a gift from Feng Zhang, Addgene plasmid #52962) was transduced with the mouse GeCKOv2 CRISPR knockout pooled library B (a gift from Feng Zhang, Addgene #1000000053), where purified virus was provided by Kevin Myant's lab[39]. Infected cells were selected with puromycin (2 μg/mL, Millipore) for 8 days. Cells were then starved for 16 h in AA-free DMEM to induce autophagy and FACS sorted to separate GFP positive and negative cells. Genomic DNA of GFP-positive cells was extracted and integrated sgRNA cassettes were amplified with Illumina primers followed by next-generation sequencing. Reads were trimmed to remove primers using cutadapt[40]. Trimmed reads from the three biological replicates of the screen were merged and Mageck (0.5.6)[41] was used to count sgRNA and perform statistical analyses with reference to a control library. Genes were ranked using the Mageck pos score statistic.

## Quantitative PCR

To measure RNA interference-mediated gene knockdown efficiency, RNA was isolated from cells using RNeasy Mini Kit (Qiagen) according to the manufacturer's instructions and cDNA synthesised from 400 ng RNA using SuperScript double stranded cDNA synthesis kit (Invitrogen). Quantitative PCR was performed using Brilliant II SYBR (Agilent Technologies) on a StepOne Plus Real-Time PCR System (Applied Biosystems). Gene expression was calculated by deltadelta Ct method and *Pld3* transcript levels were normalised to *Actin* by PCR

amplification using the following mouse primer sequences: PLD3 For 5′-ATCCATCGATGCGGTCCTTC-3′; PLD3 Rev 5′-CCAGACCAGTTGGAG GTTCC-3′; Actin For 5′-GATGAGGCYCAGAGCAAGAGAG-3′; Actin Rev 5′-GTCCCGGCCAGCCAGGTCCAG-3′.

## Immunofluorescence staining

For immunofluorescence staining, cells were plated on glass coverslips in 6-well plates for 24–48 h prior fixing with 3.7% paraformaldehyde (PFA) in 20 mM HEPES pH 7.5 for 15 min at RT. Cells were then permeabilised using 0.1% Triton X-100 in PBS for 5 min at RT or ice-cold MeOH for 2 min on ice. Blocking was then performed using 1% BSA in PBS for 30 min at RT and coverslips were then incubated with primary antibodies for 3 h at 37 °C, followed by washes in PBS and incubation with secondary Alexa fluorescence antibodies for 30 min at RT in the dark. Nuclei were subsequently stained with 4′,6-diamidino-2-pheny-lindole (DAPI; Sigma) prior to mounting on microscope slides with Prolong Gold Anti-fade (Invitrogen). Images were acquired on a Nikon A1R point scanning confocal microscope with a × 60 objective. Analyses were performed using ImageJ software. EzColocalization plugin was used to quantify Pearson's correlation coefficient (PCC) between the indicated proteins. VPS4a puncta per cell was assessed manually using a multi-point feature in ImageJ software.

## Imaging of live cells

GFP-STX17 expressing cells were plated on glass-bottom plates (World Precision Instruments) for 24 h followed by incubation in AA-free DMEM (lacking phenol red) for 2 h and subsequent addition of Lyso-Tracker Deep Red (Invitrogen L12492) or LysoTracker Red (Invitrogen L7528). Live cell imaging was performed using a Nikon A1R point scanning confocal microscope with a ×100 objective. LysoTracker ring and dot structures per cell were assessed manually using multi-point feature in ImageJ software.

## Pulldown experiments

For protein-protein interactions, HEK293T cells grown in 10-cm plates were transfected with the indicated individual constructs using PEI. 72 h post transfection, cells were harvested by direct lysing in pull-down (PD) buffer (150 mM NaCl; 20 mM Tris-HCl pH 7.5; 5 mM EDTA; 1% Triton-X) supplemented with protease inhibitors cocktail V (Fisher Scientific UK) and lysates cleared by spinning at 20,000 × g for 10 min at 4 °C. Cleared lysates expressing individual transgenes were then mixed, incubated with GFP-Trap agarose beads (Chromotek) and rotated at 4 °C overnight. Beads were then washed three times in PD buffer and proteins were eluted by boiling for 5 min in SDS sample buffer diluted in PD buffer (final concentrations: 50 mM Tris pH 6.8; 2% SDS; 10% glycerol; 1 mM 2-Mercaptoethanol). Eluted proteins were analysed by SDS-PAGE and western blot.

For unbiased analyses of protein-protein interactions, sgControl or sgRNAseK#2 U2OS cells stably expressing GFP-VPS4a<sup>WT</sup> were cultured in AA-free DMEM for 3 h. Cells were then harvested by direct lysing in PD buffer supplemented with protease inhibitors as above. Cell lysates were cleared by centrifugation at 20,000 × g for 10 min at 4 °C and incubated with GFP-Trap agarose beads (Chromotek) while rotating overnight at 4 °C. Beads were then washed three times in PD buffer followed by three washes in TBS buffer (15 mM NaCl, 10 mM Tris-HCl pH 7.5). Samples were then processed for mass spectrometry (MS) analyses as described below.

## TurboID proximity labelling

For proximity labelling experiments, cells stably expressing TurboID constructs were incubated in DMEM lacking AA supplemented with 500 μM biotin (Sigma) for 2 h. Cells were then lysed in RIPA buffer (10 mM Tris, 100 mM NaCl, 1 mM EDTA, 1 mM EGTA, 0.1% SDS, 1% Triton, 1 mM 2-Mercaptoethanol, 0.5% sodium deoxycholate, 10% glycerol) supplemented with protease inhibitor cocktail V. Samples

were centrifuged at 20,000 × g for 10 min at 4 °C and the supernatant collected and incubated with Streptavidin Sepharose High Performance beads (GE Healthcare) while rotating overnight at 4 °C. Beads were then washed twice in RIPA buffer, once in RIPA buffer containing 1 M NaCl, twice in 2 M urea in Tris-HCl (pH 8.0), and finally three washes in TBS. Samples were then processed for MS analyses as described below.

## Analyses of secreted proteins

For analyses of proteins secreted to the media, cells were incubated for 24 h in serum-free DMEM and media collected and cleared by centrifugation at 2000 × g for 20 min. Supernatants were then precipitated by the addition of 10% TCA and incubation for 30 min at 4 °C followed by centrifugation at 20,000 × g for 15 min at 4 °C. Pellets were washed in ice-cold acetone followed by centrifugation at 20,000 × g for 5 min at 4 °C. Pellets were resuspended in 6 M guanidine hydrochloride containing 1.5 mg/mL TCEP and 1 mg/mL chloroacetamide and digested with Lysyl (FUJIFILM Wako Pure Chemicals U.S.A. Corporation) for 4 h at 37 °C. Subsequently, samples were then processed for MS analyses by Trypsin digest, alkylation, and desalting as described below.

## MS analyses for pulldown, TurboID and secreted proteins

Samples were processed for MS analyses by an overnight on-bead digestion in 100 μL digest buffer with 2 μg/mL trypsin (Thermo Scientific) in proteolysis buffer (2 M urea, 50 mM Tris-HCl pH 7.5, 1 mM DTT) followed by alkylation with 10 mM Iodoacetamide and desalting using Stage-tips as previously described[42]. Peptides were resuspended in 12 μL 0.1% TFA in water. 5 μL were then injected and separated on an Ultimate 3000 Nano using a C18 packed emitter (Aurora, IonOptiks, Australia) with an increasing acetonitrile gradient, using a 40 min gradient (from 4% to 29% acetonitrile), with a 10 min 80% wash. 0.5% acetic acid was present throughout. Peptides were analysed in data-dependent mode on a Thermo Fusion Lumos with MS1 resolution 120k scanning 350–1400 and rapid ion trap MS2 scan, collision set to 30. FragPipe was used for data analysis using the pre-set "LFQ-MBR" workflow[43] and searching against the Uniprot *Mus musculus* or *Homo sapiens* databases.

## SDS-PAGE and western blotting

For western blot analyses of cultured cells, lysates were prepared by direct scraping in RIPA buffer supplemented with protease inhibitor cocktail V and cleared by centrifugation at 20,000 × g for 10 min at 4 °C. The cleared lysates were then mixed with SDS sample loading buffer (final concentrations: 50 mM Tris pH 6.8; 2% SDS; 10% glycerol; 1 mM 2-Mercaptoethanol) and heated at 95 °C for 5 min before loading on 15% or 10% acrylamide gels and SDS-PAGE separation in running buffer (Bio-Rad). Separated proteins were subsequently transferred onto nitrocellulose membranes (for 10% gels, Bio-Rad) or PVDF membranes (for 15% gels, Bio-Rad) followed by blocking in 5% non-fat milk in TBS buffer containing 0.05% TWEEN (TBST) for 30 min at RT. Subsequently, membranes were incubated with primary antibodies diluted in 5% milk in TBST overnight at 4 °C. Membranes were then washed with TBST and incubated with horseradish peroxidase-linked (HRP) secondary antibodies diluted in 5% milk TBST for 1 h at RT followed by 3 washes in TBST. Membranes were finally developed using Clarity Western ECL substrate (Bio-Rad) or SuperSignal West Femto (Thermo Scientific) to visualise protein bands in ChemiDoc XRS+ Imaging System (Bio-Rad). Densitometry analysis was performed in ImageLab software (Bio-Rad). Molecular weight markers (kDa) are indicated on the right hand side of the blots in all figures.

## Proteinase K protection assay

For proteinase K protection assay, cells were seeded in 100 mm dishes for 1 day followed by culturing in AA-free DMEM containing Baf A1

for 3 h. Subsequently, cells were washed with ice-cold PBS and harvested by direct scraping in homogenisation buffer (20 mM HEPES pH 7.6, 220 mM mannitol, 70 mM sucrose, 1 mM EDTA) followed by mechanical lysing using a 27 G needle. Samples were centrifuged at 500 *g* for 5 min at 4 °C and the supernatants collected. Each sample was then divided equally into three new tubes and the following were added: homogenisation buffer alone, Proteinase K (New England Biolabs, 25 µg/mL), or Proteinase K along with 1% Triton X-100. Samples were incubated for 10 min at 30 °C and reactions stopped by the addition of proteinase inhibitors and precipitation using 10% Trichloroacetic acid (TCA) for 30 min at 4 °C. Samples were then centrifuged 20,000 × *g* for 15 min at 4 °C. Pellets were washed in ice-cold acetone and resuspended in SDS loading buffer diluted in homogenisation buffer and heated at 95 °C for 10 min before analyses by SDS-PAGE and western blotting.

### EGFR degradation assay

For analyses of EGFR degradation rates, cells were seeded in six-well dishes for 24 h followed by incubation in serum-free DMEM for 4 h. EGFR internalisation and degradation was induced by adding 20 ng/mL EGF (PreproTech) for the indicated times and cells with lysed in RIPA buffer and analysed by SDS-PAGE and western blotting.

### Transmission electron microscopy

For ultrastructural visualisation, cells were incubated in AA-free DMEM for 3 h followed by trypsinisation and pelleting. Cell pellets were then fixed for 3 h and 15 min in fixation buffer (2.5% Glutaraldehyde, 2% Paraformaldehyde, 120 mM PIPES, 50 mM HEPES, 4 mM MgCl$_2$.6H$_2$O, 20 mM EGTA). Sections were prepared for imaging by the University of Edinburgh electron microscopy facility and visualised using Philips/FEI BioTwin CM120 Transmission Electron Microscope.

### Analyses of lysosomal function

To assess lysosomal functions, cells were seeded in 60-mm dishes for 24 h followed by incubation in full growth or AA-free DMEM for 2 h. For analyses of lysosomal acidification or Cathepsin B activity, Lysosensor Green (Invitrogen, 1 µg/mL) or Cathepsin B fluorogenic substrate III (Calbiochem, 10 µg/mL), respectively, were added to cells for 30 min. Cells were then trypsinised, resuspended in PBS, and fluorescence signal intensity measured using a BD Fortessa flow cytometer.

For ratiometric measurement of lysosomal pH, MEF cells were seeded in µ-Slide 8 Well Glass Bottom (Ibidi, 80827) and incubated for 24 h in complete media. Oregon Green 488-Dextran (Invitrogen, D7173) was added to the cells at final concentration of 0.25 mg/mL for 24 h followed by a 24 h chase. Cells were then incubated in Hanks' Balanced Salt Solution (HBSS) without phenol red (Invitrogen, 14025092). Live cell imaging was performed on a Leica Stellaris 8 confocal microscope (Leica Microsystems UK Ltd, Milton Keynes, UK) using 63X objective. Brightfield images were acquired for cell identification. To measure lysosomal pH fluorescence emission at 535 nm was measured at two different excitation wavelengths: 488 nm (pH sensitive) and 440 nm (pH insensitive). The ratio of fluorescence emissions was compared to a standard curve generated using the intracellular pH calibration buffer kit (Invitrogen, P35379) in cells labelled with Oregon Green 488-Dextran.

### Lysosomal fraction enrichment and protein content analyses

For isolation of lysosome preparations, MEF cells were treated with 10% Dextran coated magnetite (DexoMAG, 40 kDa, Liquids Research Ltd) in full growth media. 24 h later, media was replaced with fresh full growth DMEM and cells cultured for another 24 h. Cells were then pelleted and resuspended in isolation buffer (250 mM sucrose, 10 mM HEPES pH 7.4, 1 mM CaCl$_2$, 15 mM KCl, 1 mM MgCl$_2$, 1.5 mM MgAc, 1 mM DTT) followed by mechanical lysing using a 23 G needle and centrifugation at 600 × *g* for 10 min. Supernatants were loaded on a LS

magnetic separation column (Miltenyi Biotec), washed three times in isolation buffer. Magnet was then removed and bound lysosomal fractions eluted using isolation buffer followed by MS analyses as follows.

**MS sample preparation.** For lysosome-enriched fractions (ELF), protein amounts were determined using the DC protein assay (Bio-Rad) and 50 µg were precipitated by addition of 4 volumes ice-cold acetone and incubation at −20 °C overnight. Protein pellets were air dried, solubilized in freshly prepared urea digestion buffer (8 M Urea, 100 mM TEAB pH 8.5) at 37 °C for 45 min at 600 rpm. Proteins were reduced using DTT (5 mM final concentration) at 56 °C for 30 min and alkylated with acrylamide (20 mM final concentration) for 30 min at RT, followed by quenching of the reaction by addition of DTT (5 mM final concentration)[44]. Samples were diluted 1:1 with 100 mM TEAB and proteins digested with LysC (Promega) at a protease-to-protein ratio of 1:200 (w/w) overnight at 37 °C. The following day, samples were further diluted with 10 mM TEAB to a final urea concertation of 1.3 M and digested with trypsin (Promega) at a protease to protein ration of 1:50 (w/w) for 8 h at 37 °C. Subsequently, peptides were acidified by the addition of acetic acid (1% final concentration), desalted using C18 StageTips, and eluted with 80% ACN, 0.1% acetic acid[45]. Eluate fractions were dried with a vacuum centrifuge, resuspended in 5% ACN, peptide quantities were determined using the Quantitative Fluometric Peptide Assay (Thermo Fisher), and samples dried again.

**LC-MS/MS data acquisition and analysis.** Data were acquired using a Dionex Ultimate 3000 nano-UHPLC system coupled to an Orbitrap Fusion Lumos mass spectrometer (both Thermo Fisher). Analytical columns were produced in-house by packing spray tips, generated from fused silica capillaries (360 µm outer diameter, 100 µm inner diameter) with a P-2000 laser puller (Sutter Instrument), to a length of 40 cm with 3 µm ReprosilPur AC C18 particles (Dr. Maisch). Peptides were reconstituted in 5% ACN, 5% formic acid (FA) and 0.5 µg of peptides as well as indexed retention time standard (1x, Biognosis) were loaded to the analytical column with 600 nl/min solvent A (0.1% FA). Peptides were eluted with a 120 min linear gradient from 5 to 35% solvent B (90% ACN, 0.1% FA) at 300 nl/min. DIA-MS analysis parameters were adapted from a previous study[46]. *.raw files were analyzed with Spectronaut (version 14.10.201222.47784, Biognosis) using the Direct DIA approach of the Pulsar search engine (library-free DIA method). The Uniprot *M. musculus* database (63, 755 entries, release date: 03-2021) in combination with a database containing common contaminants (245 entries, release 04-2019) were used for generation of the spectral library. For database searching, the following parameters were used: enzyme specificity: trypsin/P (up to 2 missed cleavage sites); fixed modification: propionamide (Cys); variable modifications: acetylation (protein N-termini) and oxidation (Met). The most abundant 3 to 6 b/y ions (*m/z* 300–1800) per peptide were used for spectral library generation applying a 1% FDR for the identification of PSMs, peptides and proteins. The iRT-concept (deep learning assisted iRT regression) was used for retention time alignment. DIA data were matched to this library using both MS1 and MS2 spectra (maximum intensity mode) with dynamic assignment of mass tolerances for precursor and fragment ions. Default cutoffs (precursor PEP 1, precursor Qvalue 0.01, protein Qvalue 0.01 (experiment), protein Qvalue: 1 (run)) were applied for peptide and protein identification. The mean area of the three most intense fragment ions was utilized for peptide quantification (with enabled interference correction). Global median-based normalization was performed for cross run normalization and protein quantification was based on the average peptide quantity of the top three peptides (QUANT 2.0 LFQ algorithm). For determination of statistical significance, differential abundance testing (unpaired *t* test) and multiple testing correction

was performed in Perseus software. Excel was used to generate volcano dot plots throughout the manuscript.

## Plasma membrane damage assay

Plasma membrane damage assay was performed as previously described[47]. Briefly, MEF cells were plated in 6-well plates for 24 h and treated with 50 μg/mL of digitonin (Sigma, D141) diluted in complete media. After 1 min incubation at 37 °C, cells were washed and allowed to recover for 1 min at 37 °C followed by staining with 100 μg/mL PI (Sigma, P4170) diluted in complete media for 1 min at 37 °C. After PI staining, cells were fixed for 15 min in 3.7% PFA in 20 mM HEPES pH 7.5 at RT. After fixation, nuclei were stained with DAPI and images acquired on a Nikon A1R point scanning confocal microscope with a × 40 objective. Analyses were performed using ImageJ software. Number of DAPI and PI positive cells was assessed manually using a multi-point feature in ImageJ software.

## Animal studies and tissue analyses

All animal experiments were performed in accordance with protocols approved by UK Home Office guidelines. Animals were fed SDS RM1 (P) diet (SDS, DS801151G10R) ad libitum. 10-12 weeks old male C57Bl6/J mice (Charles River) were used for the delivery of AAV viruses to introduce CRISPR-Cas9-mediated gene editing of *Rnasek* in the iBAT. Mice were anesthetised with Isofluorane and a dorsal incision in the skin at the interscapular area was introduced. Each iBAT pad received four injections of 10 μL AAV containing PBS solution (Gibco) in order to deliver a total of $1 \times 10^{11}$ viral genomes per fat pad. The injections and surgery were performed in Category II cabinets while maintaining appropriate safety measures. The mice were monitored for any signs of infection and given Carprofen (Rimadyl) in drinking water for 48 h post-operation. Two weeks post injection, the animals were culled and both iBAT lobes were dissected and snap-frozen on dry ice.

For protein and DNA extraction from iBAT, 30 mg of tissue was broken down using a tissue homogeniser for 20 min at 50 oscillations per second in 500 μl of ice-cold RIPA buffer supplemented with protease inhibitor cocktail V and phosphatase inhibitors (both Fisher Scientific UK). Lysates were cleared by spinning at 20,000 × g for 10 min at 4 °C. Supernatants were collected for western blot analyses (as described above) and pellets containing DNA were extracted using DNeasy Blood and tissue kit (Qiagen; #69504). For quantitative assessment of *Rnasek* gene editing, PCR amplification of the edited region was performed using the following primer sequences: For: 5'-CCCAAATCCAGCTCCTGTCC-3' and Rev: 5'-TTCCTAAGGCTTGCG GTTCC-3'. Gel purified PCR product was submitted for sequencing using the following primer: For: 5'-CCCAAATCCAGCTCCTGTCC-3', and gene knockout efficiency was determined by analysing the sequencing results using the following online tools: https://ice. synthego.com/ and https://tide.nki.nl.

## AAV production

For AAV production, 293AAV cells were transfected with the indicated AAV vectors and packaging plasmids, including pHelper Vector (Cell Biolabs; #340202) and pAAV2-DJ/9n Vector (a gift from James M. Wilson, Addgene plasmid #112865) using PEI. Cells were harvested by incubation in 0.5 M EDTA (Invitrogen; #AM9260G) for 10 min at RT and pelleted by centrifuging at 180 × g for 5 min. Cell pellets were washed in sterile PBS (Gibco) followed by resuspension in PBS and 4 cycles of freeze-thaw on dry ice to release AAV virions. Samples were then cleared by centrifuging at 10,000 × g for 10 min and supernatants were then collected and stored at −80 °C. AAV viral titres were determined using the AAVpro Titration Kit (Takara; #6233) following manufacturer's protocol and AAV dosages were adjusted in sterile PBS in a 100 μL final volume.

## Statistics and reproducibility

All immunofluorescence analyses and western blot analyses were conducted in at least three biological replicates, unless stated otherwise. For the analysis of western blot data, actin or GAPDH were used as the loading control. Statistical analyses were performed by Graph-Pad Prism 10. Unpaired two-tailed Student's *t* test was used for comparisons between two groups, unless stated otherwise. Data are presented as the means ± SEMs. *P* value of < 0.05 is considered to specify statistical significance, *P* value for each test is shown within the figure. In the Figure legends, "*N*" specifies the number of biological independent samples.

## Reporting summary

Further information on research design is available in the Nature Portfolio Reporting Summary linked to this article.

## Data availability

The RNAseK TiD and GFP-VSP4a pulldown proteomics data generated in this study have been deposited in the ProteomeXchange database under accession code PXD042727. The protein content analyses of lysosome-enriched fractions proteomics data generated in this study have been deposited in the ProteomeXchange database under accession code PXD042079. The CRISPR/Cas9 GECKO screen data generated in this study have been deposited in the SRA database under accession code PRJNA977212. Source data are provided with this paper.

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

## Acknowledgements

We thank Kevin Myant (University of Edinburgh) for providing lenti-viruses for genome-wide CRISPR screen, Stephen Mitchell (University of Edinburgh) for EM sample preparations and help with visualisation, and Val Brunton (University of Edinburgh) for help with the animal work. All illustrations were created with BioRender.com using licences numbers (HG270K9GAM, NR270LAA1A). We are grateful to members of the Gammoh laboratory for discussions and critical reading of the manuscript. N.G. is supported by a Cancer Research UK fellowship (C52370/A21586).

## Author contributions

Conceptualisation: A.N.M. and N.G.; methodology: A.N.M., A.B., S.A., P.M., D.W., A.V.K., and N.G.; investigation: A.N.M., A.B., J.E.S., J.M., T.M., A.P.W., P.M., A.V.K., and N.G.; writing—original draft: N.G.; writing—review and editing: all authors; supervision: D.W., A.V.K., and N.G.

## Competing interests

The authors declare no competing interests.
