## [Peer Review File · Nature Communications]

The V-ATPase complex component RNaseK is required for lysosomal hydrolase delivery and autophagosome degradationREVIEWER COMMENTS

Reviewer #1 (Remarks to the Author):

In this manuscript by Makar et al., the authors performed an elegant genome-wide sgRNA library screen with the goal of identifying new critical components of the autophagy pathway. Among several known components of the autophagy pathway, they identified RNaseK as a critical protein for LC3 turnover.

I have only a few reservations regarding the first part of the manuscript. However, from the proteomic analysis of the enriched lysosomes, I cannot follow the flow of the study. The authors focussed on one lysosomal protein among many and suggested that changes in this individual protein cause the observed autophagy defects: However, final proof for this hypothesis is missing. Further on, they focus, even more indirectly, on VPS4a, which was previously suggested to play a role in PLD3 delivery to lysosomes. RNaseK modulates VPS4a, however, this remains unclear.

Several lines of evidence from the author's data support a role in the v-ATPase function (TurboID, decreased ATPase subunit is SPION enriched lysosomes), but the authors don't follow this path.

The function of PLD3 as a phospholipase is more than questionable, if the authors stick to this hypothesis, they need to support it with additional data.

There are several major flaws which are pointed out below:

Major

- The title of the manuscript seems to be misleading to me; it should at least contain the gene name RNaseK
- RNaseK was suggested to form a complex with subunits of the v-ATPase. The authors claim that it is not critical for the ATPase's function because the lysosomal pH is not altered. This is a very critical point, given that several lines of evidence in this manuscript support interaction with the v-ATPase. Therefore, proper quantification of the lysosomal pH is critical. Better, ratiometric pH-dependent dyes allow a precise pH determination and should be used in addition to LysoSensor
- A table with the proteomics data of the lysosome enrichment should be provided as a supplemental table.
- Fig. 4A: It should be mentioned in the main text that several ATPase subunits are clearly reduced in the lysosome-enriched fractions. This is hard to explain with the previous data of an unchanged lysosomal pH
- Fig. 4A: PLD3 is one of many proteins with a changed abundance in the enriched lysosomes. Even other lysosomal hydrolases are reduced (Lipa, CTSD, CTSB, SMPD1 are labeled). It is unclear why the authors are now focused on PLD3.

- The authors introduce PLD3 as a putative phospholipase, though it was shown some years ago that it is active as a lysosomal exonuclease. This was shown by independent groups. Although an activity as a phospholipase cannot be excluded, it should be explicitly discussed.
- No current data provided by the authors strengthen the role of PLD3 in autophagosomal lipid turnover.
- Figure 5E: Maybe I misunderstood this part, but the authors claim that VPS4A interacts with p62, Stx17, and Rab7 (no literature is cited). I'm not aware of such interactions; is this really physical interaction (as expected in a pulldown)?
- Any explanation why RNaseK was not identified in the pulldown mass spec approach, given the suggested interaction in Figure 5G?

Minor:

- The supplemental table should be referenced in the results part on the Illumina sequencing.
- For me, it is not clear, why the authors chose iBAT for their in vivo knockdown. This should be clarified.
- Figure 3A-C: I don't understand the Pearson quantification: For me, it doesn't make sense to compare transfected cells with untransfected and calculate the Person's coefficient of the untransfected cells (which is apparently the background of the assay)
- Control experiment for the SPION enrichment of lysosomal would be valuable (lysosomal & non-lysosomal markers, enrichment for lysosomal enzymes in lysosome-enriched fractions)

Reviewer #2 (Remarks to the Author):

Agata and others reported a new delivery system of lysosomal hydrolase PLD3 for efficient autophagy degradation. Autophagy degradation is regulated by the degradation of the inner membrane of autophagosomes. However, how the inner membrane of autophagosome is degraded is unknown, which is the focus of this study. This study reported the novel finding of PLD3 controlled by RNaseK and VPS4a, in this degradative events. Overall, the findings and conclusions are novel and move the field forward. However, there is a lack of detailed mechanistic study/data to bolster the main novel conclusions. Below are a few questions that may help to clarify some details further:

Major issues:

1. Supplementary Fig 3A showed that overexpressing RNaseK-myc did not affect autophagy. It suggests that RNaseK is not important for autophagy. This contradicts the work of this study. Also

how does this data reflect that endogenous tagging does not disrupt the function of RNaseK?
Should compare RNaseK with RNaseK-myc?

2. Can the endogenous RNaseK be detected? If so, it need to validate the function and localization of RNaseK by endogenous detecting.
3. Whether the expression level of PLD3 is affected by RNaseK knockdown?
4. How does PLD3 affect autophagy? Early steps or late steps? Is it required for the degradation of the inner membrane of autolysosomes? If so, whether this function of PLD3 depends on RNaseK or VPS4a
5. How does VPS4a accumulate on lysosomal membrane in the absence of RNaseK? Besides ALIX, GAL3, Ubiquitin is well known for the recruitment of ESCRT machinery. It may need to be texted here.
6. LLOMe induces acute lysosomal membrane damage. Testing Gal3 puncta in RNaseK knockdown cells cannot rule of the possibility that RNaseK can induce minor lysosomal permeabilization. Texting Ubiquitin or ALIX in RNaseK knockdown cells may clarify this.
7. STX17 and LysoTracker assay should be used to test the effect of PLD3 or VPS4a in the IM degradation of autolysosome.
8. How can the accumulation of VPS4A on lysosomes reduce the PLD3 in lysosomes? And how does RNaseK releases VPS4A from lysosomes? Whether the transmembrane domain of RNaseK is important for this function?

Minor issues:

1. The abstract and introduction are not well written. There is no introduction of PLD3, and no explanation how and why this study fuses on RNaseK.
2. In my opinion, Fig 2 plus Fig 1A-C should be present first, because Fig2 is a strong validation of genomic screening, Fig1C, while the Fig 1D-I are more like “supplementary” data and exclude other possibilities.

Reviewer #3 (Remarks to the Author):

This study reports the discovery of RNaseK as a novel regulator of autophagosome degradation. The authors performed a genome-wide CRISPR screening to explore genes that regulate autophagy. One of the top hits was a gene named Rnasek. The authors then conducted a series of mechanistic experiments and discovered that RNaseK is exclusively involved in the autophagy process and not in lysosomal acidification or degradation. Furthermore, the authors found that RNaseK is required

for inner autophagosomal membrane degradation and identified RNaseK's subcellular location and interaction partners. The absence of RNaseK also leads to a reduction in several lysosomal proteins, including PLD3, as well as an abnormal subcellular localization of VPS4a. PLD3 was shown to be differently trafficked and secreted into the cell medium, and it is necessary for autophagosome clearance.

This study sheds light on the functions of the RNaseK protein and provides valuable data for understanding autophagosome degradation. However, additional evidence and effort are needed to support the conclusions as well as the reproducibility of the work.

More specific comments that should be addressed:

- RNaseK was identified as a regulator of autophagy process through CRISPR screening in a mouse cell line. To confirm the discovery, a CRISPR deletion of RNaseK in a human cell line, such as 293T cells, would be required.

- I suggested including the details of all of the CRISPR screening genes in the supplemental file, such as positive selection score, fold change, etc.

- For WB results in all the Figures, the authors need to indicate the replicate times. Western blots should be quantified where possible.

- It's difficult to accept that there are substantial differences ($p < 0.0001$ or $p < 0.001$ with three replicates or three independent experiments) between the indicated groups in the bar graphs in Figs. 2G, 2J, 4C, 4G, 5B, 5C, and 5D. The authors must double-check the raw data and submit the raw data in supplemental files.

- For the significant decrease proteins identified from lysosome-enriched fractions derived from RNaseK knockout cells, the authors only studied PLD3 in depth. It's better to provide a global analysis of these proteins and explain why only focus on PLD3. In addition, there are some unlabeled dots in Fig. 4.

REVIEWER COMMENTS

Reviewer #1 (Remarks to the Author):

In this manuscript by Makar et al., the authors performed an elegant genome-wide sgRNA library screen with the goal of identifying new critical components of the autophagy pathway. Among several known components of the autophagy pathway, they identified RNaseK as a critical protein for LC3 turnover.

I have only a few reservations regarding the first part of the manuscript. However, from the proteomic analysis of the enriched lysosomes, I cannot follow the flow of the study. The authors focussed on one lysosomal protein among many and suggested that changes in this individual protein cause the observed autophagy defects: However, final proof for this hypothesis is missing. Further on, they focus, even more indirectly, on VPS4a, which was previously suggested to play a role in PLD3 delivery to lysosomes. RNaseK modulates VPS4a, however, this remains unclear. Several lines of evidence from the author's data support a role in the v-ATPase function (TurboID, decreased ATPase subunit is SPION enriched lysosomes), but the authors don't follow this path. The function of PLD3 as a phospholipase is more than questionable, if the authors stick to this hypothesis, they need to support it with additional data.

There are several major flaws which are pointed out below:

We thank this reviewer for their comments which have been addressed as detailed below.

Major

- The title of the manuscript seems to be misleading to me; it should at least contain the gene name RNaseK

As suggested by this reviewer, the manuscript title has been modified to the following: "The V-ATPase complex component RNaseK is required for lysosomal hydrolase delivery and autophagosome degradation".

- RNaseK was suggested to form a complex with subunits of the v-ATPase. The authors claim that it is not critical for the ATPase's function because the lysosomal pH is not altered. This is a very critical point, given that several lines of evidence in this manuscript support interaction with the v-ATPase. Therefore, proper quantification of the lysosomal pH is critical. Better, ratiometric pH-dependent dyes allow a precise pH determination and should be used in addition to LysoSensor

As suggested by the reviewer, we have tested a ratiometric pH-dependent dye (Oregon Green-488 dextran, PMID: 28237098) in order to compare lysosomal pH in control cells and cells lacking RNaseK. As predicted, cells lacking RNaseK exhibited no change in their lysosomal pH compared to control cells (newly added Fig. 3b). This is consistent with previous findings showing that lysosomes remain acidified in the absence of RNaseK (PMID: 26212330) as well as our data showing that the degradation of the endosomal cargo, EGFR, and the processing of Cathepsin B are intact in RNaseK-deficient cells.

- A table with the proteomics data of the lysosome enrichment should be provided as a supplemental table.

The proteomics data related to lysosome enrichment are included as Supplementary Table 3.

- Fig. 4A: It should be mentioned in the main text that several ATPase subunits are clearly reduced in the lysosome-enriched fractions. This is hard to explain with the previous data of an unchanged lysosomal pH

The significance of the reduction in ATPase subunits in the lysosomal fraction of RNAseK-deficient cells remains an open question as this does not correlate with a defect in lysosomal acidification or general degradation. This was further discussed on pg. 6 of the revised manuscript.

- Fig. 4A: PLD3 is one of many proteins with a changed abundance in the enriched lysosomes. Even other lysosomal hydrolases are reduced (Lipa, CTSD, CTSB, SMPD1 are labeled). It is unclear why the authors are now focused on PLD3.

We focused our study on PLD3 for a number of reasons (further discussed on pg. 6 of the revised manuscript). First, we were specifically interested in investigating a lysosomal lipase that was reduced in RNAseK-deficient lysosomes as we observed a defect in the inner autophagosomal degradation in the absence of RNAseK. Furthermore, PLD3 appears to be functionally similar to the sole lysosomal lipase essential for autophagy in yeast, Atg15, whereby both proteins contain an N-terminal transmembrane domain and are delivered to lysosomes through an unconventional pathway requiring VPS4. In addition, PLD3 is amongst the most highly expressed lysosomal lipases in MEFs cells where the lysosome-enrichment analyses was performed (see figure below). Finally, we further justified our focus on PLD3 by introducing CRISPR/Cas9-mediated inhibition of additional lysosomal lipases, including LIPA and SMPD1, and observed that PLD3 knockout resulted in the strongest inhibition of autophagosome degradation (newly added Supplementary Fig. 4b). The potential requirement for additional lysosomal lipases was discussed on pg. 9 of the revised manuscript. Altogether, our data show that PLD3 is an important player required for autophagosome degradation.

- The authors introduce PLD3 as a putative phospholipase, though it was shown some years ago that it is active as a lysosomal exonuclease. This was shown by independent groups. Although an activity as a phospholipase cannot be excluded, it should be explicitly discussed.

The role of PLD3 as an exonuclease has been discussed in the revised manuscript on pg. 3 and pg. 9.

- No current data provided by the authors strengthen the role of PLD3 in autophagosomal lipid turnover.

As suggested by this reviewer, to provide evidence that PLD3 can affect autophagosomal lipid turnover, we tested changes in cellular lipid content upon PLD3 knockout. As seen in the figure included below, PLD3 knockout resulted in a considerable shift in the cellular lipidome landscape. Pathway analyses of the effect of PLD3 knockout during starvation-induced autophagy show that lipid turnover is affected, particularly the reversal of the reactions $PC \rightleftharpoons LPC$, $PE \rightleftharpoons LPE$, $PE \rightleftharpoons PS$, and $PE \rightleftharpoons PC$, demonstrating the effect of PLD3 on lipid membranes. The reduction in triacylglycerides synthesis ($DG \rightleftharpoons TG$) also shows an inversion of direction, which may reflect a reduction in the recycling of fatty acids. These data indicate that PLD3 knockout can indeed influence cellular lipid turnover. We have included the data below for the reviewer's reference as we aim to expand these studies in the future to further analyse the role of PLD3 in modulating autophagosomal lipid degradation. We have expanded the discussion on pg. 9 of the revised manuscript to include the specific requirement to investigate the role of PLD3 during autophagosomal lipid turnover.

Analyses of cellular lipid content by MS comparing the effects of autophagy induction by amino acid starvation in control (sgControl) and PLD3 KO (sgPLD3) cells. Shown are pathway analyses (panels A&B) and volcano plots (panels C&D) highlighting major changes in the indicated lipids.

- Figure 5E: Maybe I misunderstood this part, but the authors claim that VPS4A interacts with p62, Stx17, and Rab7 (no literature is cited). I'm not aware of such interactions; is this really physical interaction (as expected in a pulldown)?

The VPS4A pulldown experiment presented in the above mentioned figure was performed in whole cell lysates so it is likely that many of the identified pulled-down proteins are due to a mixture of both direct and indirect protein-protein interactions. In the absence of RNAseK, VPS4a pulldown was enriched with multiple factors that are localised to late endosomes/lysosomes (including Rab7, Stx17 and p62) when compared to control cells expressing RNAseK. The text has been edited on pg. 7 to highlight the possibility that these enhanced interactions of VPS4a are due to direct or indirect protein-protein bindings as a result of stabilised localisation of VPS4a on membranous compartments.

- Any explanation why RNaseK was not identified in the pulldown mass spec approach, given the suggested interaction in Figure 5G?

RNaseK is a small protein containing only 98 amino acids. We have noticed in our experiments that even when expressed as a GFP-fusion protein followed by GFP-pulldown, we are able to detect GFP sequences but not RNaseK (despite detecting GFP-RNaseK band of the right molecular weight by western blotting). This suggests that RNaseK is difficult to detect by MS analyses, potentially due to its length as well as the location of arginine and lysine residues required for its proteolytic digestion during MS sample processing. Indeed, it is predicted that only one unique peptide of 10 amino acids can be obtained upon proteolytic digestion of RNaseK that can be detected by MS highlighting the technical difficulty of detecting RNaseK by MS analyses. A discussion of this has been included in the revised manuscript on pg. 8.

Minor:

- The supplemental table should be referenced in the results part on the Illumina sequencing.

Supplementary tables 1 & 2 that include the Illumina sequencing data have been referenced on pg. 4 of the revised manuscript.

- For me, it is not clear, why the authors chose iBAT for their in vivo knockdown. This should be clarified.

We have used the iBAT model to check the effects of knocking out RNaseK in vivo as iBAT can be transduced efficiently using AAV-based vectors (part of independent ongoing studies in the lab). The manuscript text has been edited on pg. 4 to clarify the use of the iBAT model.

- Figure 3A-C: I don't understand the Pearson quantification: For me, it doesn't make sense to compare transfected cells with untransfected and calculate the Pearson's coefficient of the untransfected cells (which is apparently the background of the assay)

As suggested by the reviewer, we have removed the graphs shown in this figure (renamed to Fig. 4a-c) and have included Pearson's coefficient values for the colocalisation of RNaseK with the indicated markers in the figure legend, thus omitting the quantifications of untransfected cells.

- Control experiment for the SPION enrichment of lysosomal would be valuable (lysosomal & non-lysosomal markers, enrichment for lysosomal enzymes in lysosome-enriched fractions)

We have included data to show the enrichment of a lysosomal marker (LAMP1) and enzyme (Cathepsin B), but not non-lysosomal markers (including EEA1 and Tubulin), in the SPION enrichment method (Supplementary Fig. 4a, manuscript text pg. 6).

Reviewer #2 (Remarks to the Author):

Agata and others reported a new delivery system of lysosomal hydrolase PLD3 for efficient

autophagy degradation. Autophagy degradation is regulated by the degradation of the inner membrane of autophagosomes. However, how the inner membrane of autophagosome is degraded is unknown, which is the focus of this study. This study reported the novel finding of PLD3 controlled by RNaseK and VPS4a, in this degradative events. Overall, the findings and conclusions are novel and move the field forward. However, there is a lack of detailed mechanistic study/data to bolster the main novel conclusions. Below are a few questions that may help to clarify some details further:
Major issues:

We thank this reviewer for their comments which have been addressed as detailed below.

1. Supplementary Fig 3A showed that overexpressing RNaseK-myc did not affect autophagy. It suggests that RNaseK is not important for autophagy. This contradicts the work of this study. Also how does this data reflect that endogenous tagging does not disrupt the function of RNaseK? Should compare RNaseK with RNaseK-myc?

In Supplementary Fig. 3a, we show the detection of endogenously tagged RNaseK with a MYC-tag, RNaseK-MYC. These experiments do not involve any overexpression systems. The text has been edited on pg. 5 to clarify this. Endogenous tagging of RNaseK was used to facilitate its detection by immunofluorescence as the antibodies we developed can only detect RNaseK by western blotting (see comment below). By showing an intact LC3 flux in these endogenously tagged cells, we conclude that RNaseK's function in autophagy is not disrupted when endogenously tagged with MYC, in contrast to the block in autophagy observed in RNaseK-deficient cells.

2. Can the endogenous RNaseK be detected? If so, it need to validate the function and localization of RNaseK by endogenous detecting.

As mentioned above, we have developed antibodies that can detect RNaseK by western blotting (Supplementary Fig. 1) but not immunofluorescence. Detection and knockout of endogenous RNaseK by western blotting were shown in the original manuscript (Fig.'s 1-3 and supplementary Fig. 1). On the other hand, endogenously tagged RNaseK-MYC was used to assess the localisation of the endogenous protein (Fig. 4a-c).

3. Whether the expression level of PLD3 is affected by RNaseK knockdown?

We assessed the levels of PLD3 in MEFs whole cell lysates by MS and observed a reduction in PLD3 levels in the absence of RNaseK reinforcing its enhanced secretion and reduced lysosomal targeting upon RNaseK knockout (Supplementary Fig. 4c).

4. How does PLD3 affect autophagy? Early steps or late steps? Is it required for the degradation of the inner membrane of autolysosomes? If so, whether this function of PLD3 depends on RNaseK or VPS4a.

We investigated the role of PLD3 in autophagy in more details as requested by this reviewer. First, we show that in PLD3-deficient cells, LC3 puncta accumulated within LAMP1-positive endosomes (newly added Supplementary Fig. 4d) suggesting that autophagosome-lysosome fusion is intact in the

absence of PLD3, as seen in RNaseK-deficient cells. This is consistent with our EM findings showing the accumulation of autolysosome structures. To assess the role of RNaseK in inner autophagosomal membrane (IAM) degradation, we assayed the formation of lysotracker ring structures that are positive for STX17, as previously described (PMID: 27885029). The newly added data in Fig. 5h show the accumulation of lysotracker rings positive for STX17 in the absence of PLD3, thus suggesting that IAM degradation is disrupted. Altogether, these data suggest that autophagosome degradation is inhibited post lysosome fusion in the absence of PLD3 as observed in cells lacking RNaseK.

Our data highlight a role for PLD3 during autophagic flux. To exhibit this function, PLD3 requires to be delivered to lysosomes. Previous data have shown that in the absence of VPS4 activity, the delivery of PLD3 to lysosomes is disrupted (PMID: 29386126). Similarly, we show that the levels of PLD3 in the lysosomes of RNaseK-deficient are also diminished. Together, these findings suggest that RNaseK and VPS4a act by facilitating the delivery of PLD3 to lysosomes to ensure its role in autophagy. A discussion of this has been included on pg. 8-9 of the revised manuscript.

5. How does VPS4a accumulate on lysosomal membrane in the absence of RNaseK? Besides ALIX, GAL3, Ubiquitin is well known for the recruitment of ESCRT machinery. It may need to be texted here.

As suggested by this reviewer in this point and subsequent one, we tested the localisation of ALIX and Ubiquitin in cells lacking RNaseK. The data added as Supplementary Fig. 5c show that RNaseK-deficient cells did not accumulate ALIX puncta unlike cells treated with LLOMe further reinforcing that the absence of RNaseK in cells does not elicit lysosomal perturbations.

On the other hand, when RNaseK-deficient cells were stained with an antibody to detect endogenous ubiquitin, we observed enhanced Ubiquitin puncta formation that localised within lysosomes (see figure below). These Ubiquitin puncta were not observed in control cells or control cells treated with LLOMe. We reasoned that this enhanced Ubiquitin puncta within lysosomes in the absence of RNaseK may be due to the accumulation of undegraded cargo, including ubiquitinated substrates. Indeed, staining control cells treated with Baf A1 also resulted in the accumulation of lysosome-localised ubiquitin puncta (see figure below). These findings, added below for the reviewer's reference, suggest that the accumulation of ubiquitinated proteins within lysosomes in RNaseK-deficient cells are potentially due to inhibited cargo degradation in the absence of RNaseK.

Analyses of ubiquitin localization in sgControl or sgRNAseK cells. Ubiquitin puncta localized within lysosomes are only seen in sgRNAseK cells or sgControl cells treated with Baf A1 (2 h treatment) to inhibit lysosomal degradation. On the other hand, no ubiquitin puncta accumulation was observed in lysosomes of sgControl cells or sgControl cells treated with LLOMe. Quantifications of ubiquitin-lysosome co-localisation mean values from at three independent experiments are shown below. Mean + SEM values are shown. **** p<0.0001, assessed by unpaired Student's t-test.

6. LLOMe induces acute lysosomal membrane damage. Testing Gal3 puncta in RNaseK knockdown cells cannot rule out the possibility that RNaseK can induce minor lysosomal permeabilization. Testing Ubiquitin or ALIX in RNaseK knockdown cells may clarify this.

This point has been addressed in the previous one (point 5) by the same reviewer.

7. STX17 and LysoTracker assay should be used to test the effect of PLD3 or VPS4a in the IM degradation of autolysosome.

We have tested the formation of lysotracker rings positive for STX17 in PLD3-deficient cells in point 4 above and show that the degradation of the inner autophagosomal membrane is disrupted (newly added Fig. 5h).

8. How can the accumulation of VPS4A on lysosomes reduce the PLD3 in lysosomes? And how does RNaseK release VPS4A from lysosomes? Whether the transmembrane domain of RNaseK is important for this function?

These are interesting questions raised by the reviewer. To address them, we have performed the following.

First, previous data have shown that VPS4 activity is required for the efficient recruitment of PLD3 to lysosomes (PMID: 29386126) through an unconventional delivery pathway. As VPS4 is usually found diffused in cells, we predict that by being arrested on lysosomes in RNaseK-deficient cells, some of its activities are inhibited. This is also supported by the observation that the dominant negative mutant of VPS4a (VPS4a^{E228Q}) is mostly localised to punctate structures (Fig. 6i). To test this further, we investigated an independent cellular pathway in cells lacking RNaseK by analysing plasma membrane repair which has been previously shown to require the activity of VPS4 (PMID: 24482116) but not core autophagy components (PMID: 30478389). The newly added data in Supplementary Fig. 5d show that digitonin-induced plasma membrane repair was defective in cells lacking RNaseK thus providing a functional assay for an alternative cellular pathway requiring VPS4 activity. On the other hand, how RNaseK can regulate VPS4 activity and release from lysosomes remains an open question and has been further discussed on pg. 9 of the revised manuscript.

Finally, we predict that the transmembrane domain of RNaseK is essential for its proper localisation to lysosomes and therefore activity. However, testing the role of the transmembrane domain of RNaseK in binding or regulating VPS4a activity requires introducing large deletion mutants as RNaseK is only 98 amino acids in length and harbours two transmembrane domains (see depiction below). Such mutations are unlikely to result in proper protein expression and folding as we have observed upon introducing large mutations in RNaseK (data not shown).

A schematic diagram of RNaseK highlighting the transmembrane regions. Protter software was used for protein annotation and structural prediction.

Minor issues:

1. The abstract and introduction are not well written. There is no introduction of PLD3, and no explanation how and why this study focuses on RNaseK.

We have modified the abstract and introduction to include information on PLD3 and how we identified RNaseK.

2. In my opinion, Fig 2 plus Fig 1A-C should be present first, because Fig2 is a strong validation of genomic screening, Fig1C, while the Fig 1D-I are more like “supplementary” data and exclude other possibilities.

As suggested by this reviewer, we have moved figures 1 and 2 and split new figure 1 into two figures so that they included the validation of the screen and new figure 3 the effects of RNaseK on lysosomal activity.

Reviewer #3 (Remarks to the Author):

This study reports the discovery of RNaseK as a novel regulator of autophagosome degradation. The authors performed a genome-wide CRISPR screening to explore genes that regulate autophagy. One of the top hits was a gene named Rnasek. The authors then conducted a series of mechanistic experiments and discovered that RNaseK is exclusively involved in the autophagy process and not in lysosomal acidification or degradation. Furthermore, the authors found that RNaseK is required for inner autophagosomal membrane degradation and identified RNaseK's subcellular location and interaction partners. The absence of RNaseK also leads to a reduction in several lysosomal proteins, including PLD3, as well as an abnormal subcellular localization of VPS4a. PLD3 was shown to be differently trafficked and secreted into the cell medium, and it is necessary for autophagosome clearance.

This study sheds light on the functions of the RNaseK protein and provides valuable data for understanding autophagosome degradation. However, additional evidence and effort are needed to support the conclusions as well as the reproducibility of the work.

We thank this reviewer for their comments which have been addressed as detailed below.

More specific comments that should be addressed:

- RNaseK was identified as a regulator of autophagy process through CRISPR screening in a mouse cell line. To confirm the discovery, a CRISPR deletion of RNaseK in a human cell line, such as 293T cells, would be required.

As suggested by this reviewer, we have included data to show the effects of deleting RNaseK in the human osteosarcoma cell line, U2OS. The newly added data in Supplementary Fig. 1b show that RNaseK can regulate autophagy in human cells as seen in mouse cells.

- I suggested including the details of all of the CRISPR screening genes in the supplemental file, such as positive selection score, fold change, etc.

A detailed supplemental file has been added including more details of the CRISPR screening results, such as positive selection score and p-values (Supplementary Table 2, cited on pg. 4).

- For WB results in all the Figures, the authors need to indicate the replicate times. Western blots should be quantified where possible.

As suggested by this reviewer, we have included additional western blot quantifications (e.g. figures 1f, 3d, 6g, 6h and supplementary figures 1b, 4b) and ensured that replicate times is specified in the figure legends. Quantifications of the following WB results were included in the first submission (e.g. figures 1e, 1i, 3h, 5i, supplementary figure 2c).

- It's difficult to accept that there are substantial differences ($p < 0.0001$ or $p < 0.001$ with three replicates or three independent experiments) between the indicated groups in the bar graphs in Figs. 2G, 2J, 4C, 4G, 5B, 5C, and 5D. The authors must double-check the raw data and submit the raw data in supplemental files.

We have double checked the statistical analyses of all experiments and adjusted values as appropriate. In some of the graphs, we have shown SD values instead of SEM values. This has been corrected throughout the manuscript. We thank the reviewer for pointing this out. None of the introduced modifications changed any of the conclusions in this manuscript.

The raw data has been uploaded as per editorial requirements.

- For the significant decrease proteins identified from lysosome-enriched fractions derived from RNAseK knockout cells, the authors only studied PLD3 in depth. It's better to provide a global analysis of these proteins and explain why only focus on PLD3. In addition, there are some unlabeled dots in Fig. 4.

The rationale behind focusing on PLD3 in depth has been addressed above (reviewer 1, point 5).

We have also uploaded a supplementary file (Supplementary Table 3) to include all the datapoints in the volcano plot as we are unable to label all dots on the graph.

REVIEWER COMMENTS

Reviewer #1 (Remarks to the Author):

In this revised version of the manuscript by Makar et al., the authors addressed some of my concerns but not all.

There are still some inconsistencies in the datasets presented:

- how could it be that the lysosomal pH is not changed despite the reduced levels of (several!) v-ATPase subunits?

- how can the authors be sure that the deficits in autophagy caused by RNaseK deficiency are due to lowered PLD3 levels in lysosomes? Couldn't it just be the sum of all proteomic changes, lack of proteases, or general lysosomal dysfunction? Phenocopy of the upon PLD3 knockdown is evidence, but there is no proof.

A major point in this manuscript is the hypothesis that PLD3 acts as a Phospholipase D, thereby degrading lipids. The argument „ PLD3 appears to be functionally similar to the sole lysosomal lipase essential for autophagy in yeast, Atg15, whereby both proteins contain an N-terminal transmembrane domain and are delivered to lysosomes through an unconventional pathway requiring VPS4.“... is not a very strong argument for this presumed lipolytic activity.

Showing compelling evidence for its lipid-degrading activity is critical. Though the authors present a lipidomics analysis (apparently only for the reviewers but not for the published manuscript), this still does not indicate direct hydrolytic activity of PLD3 towards lipids as substrates but could be secondary. Which lipids can be degraded by PLD3? By far, not all lipids changed in the volcano plot are typical (putative) substrates for PLD enzymes. These questions need robust experimental testing of their enzymatic function, especially because the lipolytic activity was challenged by several groups.

Given the importance for the story, this is a critical point!

For me, the link between VPS4a, RNaseK, and PLD3 still remains dubious. It should be noted that the authors of the cited reference (Reference 10; Gonzalez, A. C. et al. Unconventional Trafficking of Mammalian Phospholipase D3 to Lysosomes. Cell Rep 22, 1040-1053 (2018)) don't claim a direct interaction between PLD3 and VPS4a, but instead, an MVB-dependent delivery that critically depends on VPS4a as an essential component of the MVB sorting machinery. Is the MVB pathway, in general, impaired in RNaseK –depleted cells? And if so, wouldn't that explain the effect of impaired autophagy?

Reviewer #2 (Remarks to the Author):

The authors have successfully addressed all raised questions and the manuscript is now significantly improved

Reviewer #3 (Remarks to the Author):

The authors have done a great job in responding to reviewer critiques, including new data, raw data, quantifications, and discussion. I have no new or remaining suggestions that would be within the scope of this publication.

REVIEWER COMMENTS

Reviewer #1 (Remarks to the Author):

In this revised version of the manuscript by Makar et al., the authors addressed some of my concerns but not all.

There are still some inconsistencies in the datasets presented:

- how could it be that the lysosomal pH is not changed despite the reduced levels of (several!) v-ATPase subunits?

This is indeed an interesting observation highlighted by the reviewer. We have experimentally shown that the lysosomal pH is not affected by the absence of RNaseK (Figure 3), consistent with previous publications (PMID: 26212330 and PMID: 27776355). It is possible that the reduction in V-ATPase levels is not sufficient to cause a disruption in proton pumping activity. This has been discussed on pg.6 of the revised manuscript.

- how can the authors be sure that the deficits in autophagy caused by RNaseK deficiency are due to lowered PLD3 levels in lysosomes? Couldn't it just be the sum of all proteomic changes, lack of proteases, or general lysosomal dysfunction? Phenocopy of the upon PLD3 knockdown is evidence, but there is no proof.

We agree with this reviewer that multiple factors may contribute to the defect in autophagosome degradation in the absence of RNaseK. Indeed, knockdown of lysosomal lipases reduced in the absence of RNaseK also resulted in inhibited autophagy to varying degrees, albeit to lower extent compared to PLD3 knockdown (supplementary Figure 4b). This has been discussed on pg.9 of the manuscript (last paragraph of the discussion). We have modified the manuscript to further highlight this point (pg.7 and pg.8). On the other hand, lysosomal degradation of the endosomal cargo EGFR remained intact in RNaseK-deficient cells suggesting that there is no general lysosomal dysfunction that contributes to autophagy inhibition.

A major point in this manuscript is the hypothesis that PLD3 acts as a Phospholipase D, thereby degrading lipids. The argument „ PLD3 appears to be functionally similar to the sole lysosomal lipase essential for autophagy in yeast, Atg15, whereby both proteins contain an N-terminal transmembrane domain and are delivered to lysosomes through an unconventional pathway requiring VPS4.“... is not a very strong argument for this presumed lipolytic activity. Showing compelling evidence for its lipid-degrading activity is critical. Though the authors present a lipidomics analysis (apparently only for the reviewers but not for the published manuscript), this still does not indicate direct hydrolytic activity of PLD3 towards lipids as substrates but could be secondary. Which lipids can be degraded by PLD3? By far, not all lipids changed in the volcano plot are typical (putative) substrates for PLD enzymes. These questions need robust experimental testing of their enzymatic function, especially because the lipolytic activity was challenged by several groups.

Given the importance for the story, this is a critical point!

As mentioned by this reviewer and stated in our manuscript, the phospholipase activity of PLD3 is disputable. We therefore sought to test whether PLD3 harbours a lipid modifying capacity. To do so, we used purified PLD3 protein and an Amplex Red Phospholipase D Assay Kit and observed that

PLD3 harbours a detectable phospholipase activity when using phosphatidylcholine as a substrate (see figure below). However, these results require further analyses and the inclusion of additional controls to ensure enzymatic specificity. Therefore, to address this concern, as well as the above point, we have modified the conclusions of this manuscript to clarify that the phospholipase activity of PLD3 is uncertain and that multiple factors may contribute to the disruption of autophagy in the absence of RNAseK. These modifications have been highlighted throughout the text.

Potential PLD3 phospholipase activity was tested using Amplex Red Phospholipase D assay (Invitrogen, A12219), according to the manufacturer's instructions. Briefly, recombinant PLD3 or BSA control were incubated with phosphatidylcholine substrate and phospholipase activity was measured as an increase in Amplex Red fluorescence. Shown are raw fluorescence signal values. **** $p < 0.0001$, (ns) non-significant assessed by paired Student's t-test. N=3.

For me, the link between VPS4a, RNAseK, and PLD3 still remains dubious. It should be noted that the authors of the cited reference (Reference 10; Gonzalez, A. C. et al. Unconventional Trafficking of Mammalian Phospholipase D3 to Lysosomes. Cell Rep 22, 1040-1053 (2018)) don't claim a direct interaction between PLD3 and VPS4a, but instead, an MVB-dependent delivery that critically depends on VPS4a as an essential component of the MVB sorting machinery. Is the MVB pathway, in general, impaired in RNAseK-depleted cells? And if so, wouldn't that explain the effect of impaired autophagy?

We have focused on investigating lysosome biology as the disruption in autophagy in RNAseK knockout cells occurs after lysosome fusion. Indeed, a potential defect in the MVB pathway as a result of VPS4a mislocalisation can lead to inhibited delivery of lysosomal hydrolases, in agreement with our findings. We have modified the text on pg.8 to clarify this.

REVIEWERS' COMMENTS

Reviewer #1 (Remarks to the Author):

All my points were discussed /partially addressed experimentally.

I think if the speculated lipid-degrading effect towards the autophagosomal membrane had worked out better, this would have been very relevant.